# Dissecting the immune suppressive human prostate tumor microenvironment via integrated single-cell and spatial transcriptomic analyses

Taghreed Hirz [1,2,3,12] ✉, Shenglin Mei [1,4,12] ✉, Hirak Sarkar[4], Youmna Kfoury[1,2,3], Shulin Wu[5,6], Bronte M. Verhoeven[7], Alexander O. Subtelny[5], Dimitar V. Zlatev[6], Matthew W. Wszolek[6], Keyan Salari [6,8], Evan Murray[8], Fei Chen [8], Evan Z. Macosko [8,9], Chin-Lee Wu[5,6], David T. Scadden[1,2,3], Douglas M. Dahl[6], Ninib Baryawno [7,12], Philip J. Saylor [10,12], Peter V. Kharchenko [2,4,8,11,12] & David B. Sykes [1,2,3,12] ✉

The treatment of low-risk primary prostate cancer entails active surveillance only, while high-risk disease requires multimodal treatment including surgery, radiation therapy, and hormonal therapy. Recurrence and development of metastatic disease remains a clinical problem, without a clear understanding of what drives immune escape and tumor progression. Here, we comprehensively describe the tumor microenvironment of localized prostate cancer in comparison with adjacent normal samples and healthy controls. Single-cell RNA sequencing and high-resolution spatial transcriptomic analyses reveal tumor context dependent changes in gene expression. Our data indicate that an immune suppressive tumor microenvironment associates with suppressive myeloid populations and exhausted T-cells, in addition to high stromal angiogenic activity. We infer cell-to-cell relationships from high throughput ligand-receptor interaction measurements within undissociated tissue sections. Our work thus provides a highly detailed and comprehensive resource of the prostate tumor microenvironment as well as tumor-stromal cell interactions.

Localized prostate cancer (PCa) is a clinically heterogeneous disease. Some patients present with low-risk prostate tumors that can safely be observed, while others have a high-risk disease that carries a substantial relapse risk even following state-of-the-art treatment. Despite efforts aimed at early detection and improving our current curative-intent therapies, many patients, unfortunately, experience recurrence[1]. There remains a significant need to further our understanding of PCa, where biological insights of the prostate tumor microenvironment (TME) may help to identify novel therapeutic targets.

Single-cell gene expression technologies have made it possible to assess thousands of cells within a single sample, revealing subtleties in tumor cell heterogeneity as well as a complex TME[2,3]. Examinations of normal adult human prostate[4] and PCa have provided detailed descriptions of the epithelial and tumor cells as well as cell states in both prostate adenocarcinoma[5–8] and neuroendocrine tumors[9]. However, the immune cells within the prostate microenvironment have not been rigorously characterized at the single-cell level. The prostate TME typically contains few immune cells, and it is hypothesized that this

feature may explain the generally poor response of PCa to immunotherapy[10]. We, therefore, processed freshly collected prostate tissues using a method that enriched and preserved immune cells to characterize the immune microenvironment at high-resolution.

To validate our single-cell findings, we used a spatial transcriptomic technique (Slide-seqV2), where the tissue architecture and cell-cell proximity relationships are preserved[11,12]. We developed a new computational means of data analysis to examine the transcriptional impact of tumor cells on neighboring stromal cells, including fibroblasts, pericytes, and endothelial cells.

In this study, combined scRNA-seq and spatial transcriptomic analyses improve our understanding of PCa with the following observations: (1) primary PCa establishes a suppressive immune microenvironment, (2) the prostate TME exhibits a high angiogenic gene expression pattern in addition to a (3) new computational analysis pipeline to deconvolute context-specific differential gene expression. We further reveal the transcriptional state of stromal cells based on their spatial localization within the tumor. In sum, our data reveal a highly immune-suppressive TME and describe tumor-induced alterations of neighboring cells that promote tumorigenesis and progression. This careful dissection of the cellular and molecular landscape of PCa will help identify areas of vulnerability amenable to therapeutic intervention.

## Results

### The prostate TME characterized by single-cell and spatial transcriptomic analysis

Fresh PCa samples were collected from 19 treatment-naive patients diagnosed with prostate adenocarcinoma and underwent radical prostatectomy. In 15 out of the 19 patients, matched 'normal' benign prostate tissue adjacent to the tumor was also sampled. As controls, prostate tissues not harboring cancer were collected from 4 patients (underwent cystoprostatectomy for bladder cancer), and one healthy prostate tissue was collected as part of a rapid autopsy from a patient with metastatic non-small cell lung cancer (Fig. 1a).

The cellular composition of the prostate TME was examined across a spectrum of primary tumor grades and stages (pathologic T-stage 2a to 3b; Gleason score 6-10). Samples were divided into low-grade (LG, Gleason 6 and 7, 12 cases) and high-grade (HG, Gleason 8–10, 7 cases) (Supplementary Data 1). Live, nonerythroid cells (DAPI[neg]/CD235[neg]) were collected by fluorescence-activated cell sorting (FACS) from healthy prostate tissues ($n = 5$), prostate tumor tissues ($n = 12$ LG and $n = 7$ HG), and adjacent non-tumor prostate tissues ($n = 11$ LG and $n = 4$ HG, hereafter 'adjacent-normal' (Adj-N)). All patients had standard pathologic evaluation to confirm their diagnosis (Fig. S1a).

The transcriptomes of 179,359 single cells were analyzed (average of 4721 cells per sample and 50,416 transcripts per cell, Supplementary Data 2). Conos[13] (Clustering On Network Of Samples) aligned the samples, and the analysis of the resulting joint cell clusters revealed a rich repertoire of immune cells and non-immune stromal cells (Fig. 1b). Cell types were annotated based on cell type-specific gene markers, forming 16 major clusters (Fig. 1c, Fig. S1b, Supplementary Data 3).

Of note, our dissociation protocol was optimized to enrich immune cells. This was an intentional choice to focus on the prostate immune TME with the goal of understanding why PCa is considered poorly immunogenic and rarely respond to immunotherapy[14]. In comparing our tissue processing method (Collagenases+Dispase) to a published protocol of a single-cell prostate study (Rocky)[4], the Collagenases+Dispase released a higher proportion of immune cells (Fig. 1d, Fig. S1c). Reassuringly, cells liberated by both dissociation protocols showed similar transcriptome profiles (Fig. S1d).

In terms of the abundance of major cell populations, significant but small differences were observed in plasma cells, macrophages, and endothelial cells when comparing the tumor fraction to the Adj-N

(Fig. 1e). Stratifying LG and HG cases, there were similarly small but significant changes in plasma cells (Adj-N vs. tumor, LG), macrophages (Adj-N vs. tumor, LG) and endothelial cells (Healthy vs. adj-N LG) (Fig. S1e). The few significant differences in cell abundance were likely due to high patient-to-patient variability even within patients with same Gleason score (Fig. S1f).

The overall similarity of the transcriptional state between samples was examined using a weighted expression distance, revealing a significant increase in the inter-patient variability among the tumor fraction, compared to the adj-normal and healthy fractions (Fig. 1f). This suggests divergent trajectories of the cellular states in the tumor region among different patients.

To validate single-cell findings with a dissociation-free approach that preserves tissue architecture, we performed spatial transcriptomics using Slide-seqV2[11,12]. This provided the opportunity to examine tumor organization at high spatial resolution. Fresh-frozen 10-micron sections were sampled from two healthy prostate samples and two prostate tumor samples (one of LG Gleason score and one of HG Gleason score) as well as their corresponding adjacent-normal tissues (Fig. 1g).

Robust Cell Type Decomposition (RCTD) was used to assign cell type annotations on Slide-seqV2 data based on scRNA-seq reference data (see 'Methods')[15]. Hallmark genes denoting different cell populations were used to verify the RCTD annotation (Fig. S1g). As expected, Slide-seqV2 measurements showed more pronounced differences in cell proportions as compared to the scRNA-seq data, with greatly expanded epithelial and fibroblasts populations and a significantly smaller fraction of immune cells (Fig. 1d).

The cellular architecture viewed through the lens of Slide-seqV2 was reassuringly consistent with what one would expect from standard H&E staining. The highly detailed spatial configuration of the healthy prostate tissue demonstrated well-organized prostate epithelial glands surrounded by immune and non-immune stromal cells including fibroblasts, pericytes, and endothelial cells (Fig. 1g, panel 1). This architecture was notably disrupted in the cancerous prostate (Fig. 1g, panels 3 and 4). Differences in tissue organization were quantified by spatial autocorrelation using Moran's I score, which evaluates the extent to which the cells are clustered (high score) or dispersed (low score)[16]. The Moran's I score for fibroblasts, endothelial cells, and pericytes significantly decreased in tumors as compared to healthy tissues (Fig. 1h).

### A prostate tumor gene signature distinguishes normal and malignant luminal epithelial cells

Unsupervised clustering revealed four epithelial subpopulations: basal, luminal, club, and hillock (Fig. 2a) as denoted by key-marker gene expression (Fig. S2a). Hillock and club cells were identified as transitional cells in a cellular atlas of the mouse lung[17]. These cells have also been reported in human prostate tissue[4,18] and in benign human prostate organoids[6], but their role in prostate tumorigenesis remains unclear.

We used RNA velocity to infer the likely trajectories of epithelial cell differentiation[19]. One trajectory suggested that club cells act as luminal cell progenitors, an observation previously reported in PCa[20]. A second distinct trajectory showed consistent directional flow suggesting that hillock cells may act as progenitors for basal cells (Fig. 2b). Differential gene expression comparing healthy and tumor-associated hillock and club cells showed enrichment in genes involved in urogenital system development and epithelial tubes morphogenesis, respectively (Fig. S2b), and these cells are known to be enriched in urethra and peri-urethral prostate zones[4].

Malignant cells did not cluster separately from the non-malignant epithelial populations from which they originated. To distinguish malignant cells from normal epithelial cells within the prostate tumor samples, we applied inferCNV[21] on the four epithelial subpopulations,

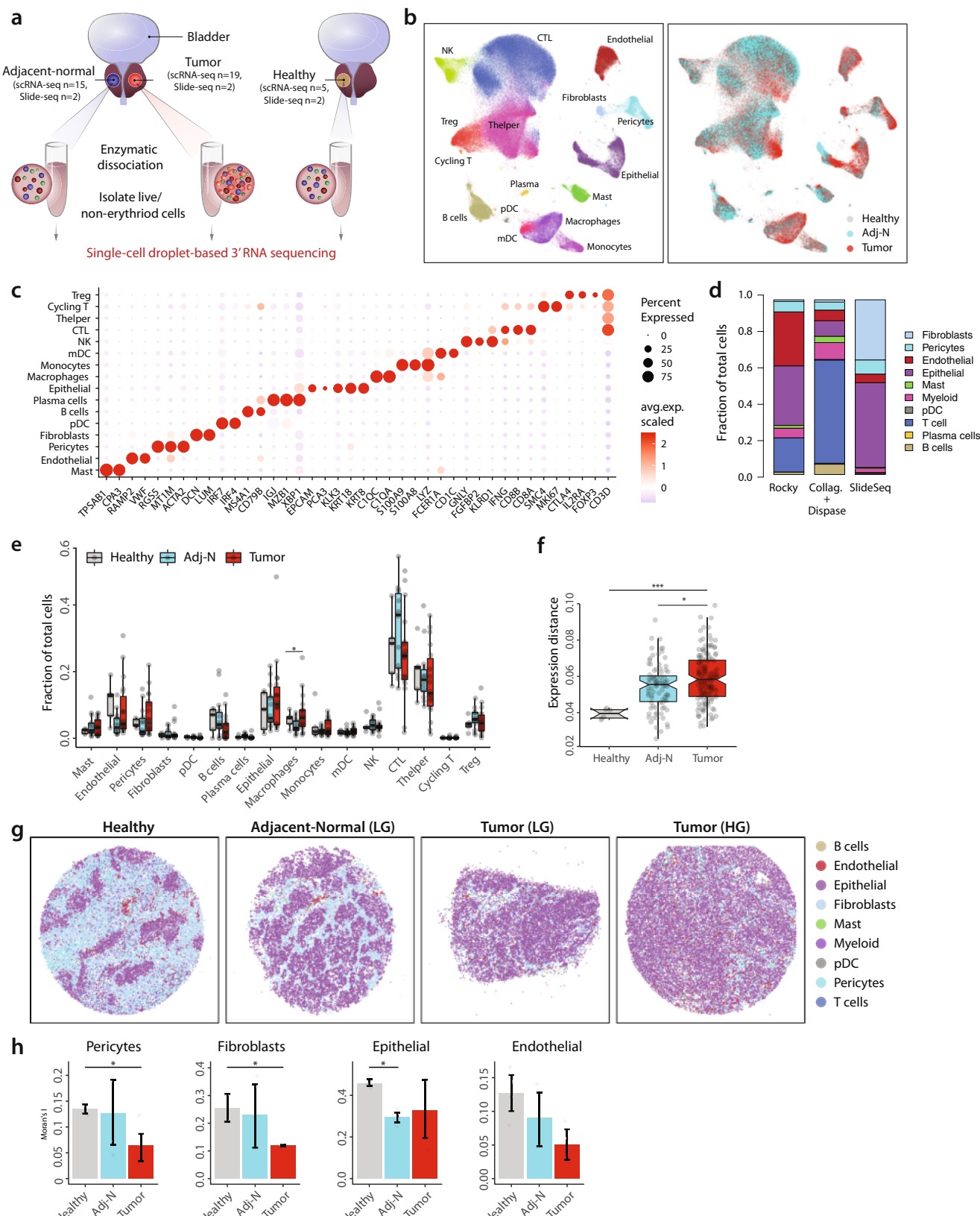

taking their corresponding subpopulation from healthy samples as a reference. Only cells within the luminal subpopulation showed clear chromosomal aberrations, indicating that the malignant cells are of luminal origin, consistent with previous studies[22] (Fig. S2c).

Chromosomal aberrations and inferCNV analysis allowed us to separate malignant luminal cells (with genomic aberrations) from normal luminal cells within the tumor. Differential expression genes (DEGs) analysis was used to identify an expression signature for the malignant cells, leading to a signature composed of eight genes, which we termed the "Prostate Tumor Gene Signature" (Fig. 2c) (See 'Methods'). We applied this gene signature to published bulk RNA-seq of PCa, demonstrating a consistent ability to distinguish tumor samples

**Fig. 1 | The prostate TME characterized by single-cell and spatial transcriptomic analyses. a** Schematic illustration of samples collection and processing. **b** Integrative analysis of scRNA-seq samples visualized using a common UMAP embedding for cell annotation (left) and sample fractions (right). **c** Dotplot representing key-marker gene expression in major cell types. The color represents scaled average expression of marker genes in each cell type, and the size indicates the proportion of cells expressing marker genes. **d** Stacked barplots showing the fractional composition of cell number for different clusters within scRNA-seq (using two different dissociation protocols: Collagenases+Dispase and Rocky, see text) and Slide-seqV2. **e** Boxplot comparing proportion of major cell populations between healthy prostate tissues ($n = 5$) and tissues collected from cancerous prostates (tumor $n = 18$ and adj-normal $n = 14$). Significance was assessed using two-sided Wilcoxon rank-sum test (Macrophage: *$p = 0.03$). **f** Boxplot showing inter-individual gene expression distances (based on Pearson correlation) within healthy, adj-normal, and tumor samples, averaged across all cell types. Significance was assessed using two-sided Wilcoxon rank-sum test (tumor vs. adj-normal *$p = 0.015$;

Tumor vs. Healthy ***$p = 0.0003$). Boxplots in **e**, **f** include centerline, median; box limits, upper and lower quartiles; and whiskers are highest and lowest values no greater than 1.5× interquartile range. **g** Spatial presentation at a high-resolution level using Slide-seqV2 for the major cell populations in healthy ($n = 4$), adj-normal of LG case ($n = 2$), and two tumor tissues collected from a low-grade (Tumor-LG $n = 2$) and high-grade (Tumor-HG $n = 2$) patients. Patinets ID from Supplementary Data 2 represented here as healthy is HP1, adj-normal of LG case is Benign04, tumor tissue of LG case is Tumor08, tumor tissue of HG case is Tumor02. **h** Barplots showing spatial autocorrelation (Moran's I) of fibroblasts and pericytes in Healthy ($n = 4$), adj-Nomral ($n = 4$), and Tumor samples ($n = 4$). Moran's I evaluates whether the cells are clustered (high Moran's I score) or dispersed (low Moran's I score). Statistical analysis was performed using two-sided Wilcoxon rank-sum test. (Pericytes *$p = 0.03$; Fibroblasts *$p = 0.029$; Epithelial *$p = 0.03$, error bars: SEM). Source data are provided as a Source Data file. *P* values <0.05 were considered significant: *$p < 0.05$; **$p < 0.01$; ***$p < 0.001$; ****$p < 0.0001$.

---

from adjacent-normal samples across four independent datasets (Fig. S2d)[23–26].

Since we were able to distinguish malignant cells from normal luminal epithelial cells within tumor samples, we assessed for malignant cells heterogeneity. Clustering of malignant cells revealed three major aspects of malignant clusters (Fig. S3a, see 'Methods'). Gene Ontology (GO) pathway analysis showed enrichment in genes related to cell growth and migration in malignant cluster 1 (C1) (Fig. S3b). C1 also showed high expression of *EGR1*, *IER2*, and *KLF6* genes (Fig. S3a) suggesting roles in PCa progression, motility, and metastasis[27,28].

Epithelial-mesenchymal transition (EMT) plays an important role in prostate cancer progression and metastasis[29]. Malignant cells showed significantly higher EMT gene signature[30] (Supplementary Data 4) as compared to non-malignant luminal epithelial cells from the three different sample types (healthy, Adj-N, and tumor) (Fig. 2d, Fig. S3c).

Spatially, the healthy prostate tissue demonstrated an organized glandular epithelium with a well-structured bilayer of basal and luminal cells (Fig. 2e). The adj-normal sample differed with an expansion of the luminal epithelial population, and loss of the well-organized glands (Fig. S1a, Fig. 2e). Epithelial subpopulations were annotated using RCTD and validated using key-marker genes for the four different epithelial subpopulations (Fig. 2e and 2f). The clusters of club and hillock cells observed within the healthy prostate tissue were disrupted in the tumor and adj-normal tissues as demonstrated by spatial autocorrelation (Fig. 2e and S3d).

The "Prostate Tumor Gene Signature" obtained from the 10X single-cell data was applied to evaluate tumor cell annotation in the Slide-seqV2 data. This eight-gene tumor signature successfully identified tumor-enriched area within the HG case (Fig. 2g). Almost no tumor cells were annotated in the healthy and adj-normal samples (Fig. 2g), speaking to the accuracy of this "Prostate Tumor Gene Signature".

### Context-dependent differential expression with linear admixture correction

Using our Slide-seqV2 data, the edge or boundary of the expanding tumor was particularly evident in the HG sample, which could be segmented into two distinct spatial contexts. The tumor context was dominated by a dense accumulation of tumor cells, while the tumor-adjacent context was composed primarily of non-malignant epithelial cells (Fig. S3e). The small fraction of tumor cells detected within the adj-normal sample likely represents real infiltration of tumor cells. Slide-seqV2 allows one to examine the differences in cellular state associated with precise spatial contexts. Annotation tools such as RCTD[15] estimate the fractions of cell types contributing to each bead and identify relatively pure beads that can be confidently assigned to a specific cell type. However, even "pure" beads can carry admixture of transcriptomes from the neighboring cells (Fig. 2h).

As composition of the cellular neighborhoods varies between different tissue contexts, such admixture will heavily bias transcriptional comparisons of cellular state between contexts. To overcome this admixture effect, we developed a new computational approach which regressed out context-dependent differences that could be attributed to admixture from other cell types, focusing on the residual differences that likely reflect the context-dependent change in the transcriptional state of the target-cell type (details in Supplementary Note). In subsequent sections, we apply this approach to contrast the state of the stromal populations between tumor and tumor-adjacent contexts.

### The prostate tumor microenvironment exhibits high endothelial angiogenic activity

The non-immune stromal populations including fibroblasts, endothelial cells, and pericytes, represent important components of the TME whose function and abundance varies significantly between cancer types[31]. We identified five stromal subpopulations including two endothelial, two pericytes, and one fibroblasts subpopulation (Fig. 3a) annotated based on key-marker gene expression (Fig. S4a and Supplementary Data 3).

Endothelial-1 subpopulation showed high expression of *SELE/SELP/CLU/PLVAP*, characteristic of sinusoidal endothelial cells whereas Endothelial-2 subpopulation expressed common arterial genes (HEY1/IGFBP3/FBLN5)[32–35] (Fig. S4a). GO analysis of Endothelial-2 pointed to pathways involved in blood vessel development and angiogenesis (Fig. 3b). An angiogenesis gene signature[30] (Supplementary Data 4) demonstrated that the tumor-associated Endothelial-2 scored highest when compared to the other stromal populations, and when comparing healthy and tumor fractions across almost all non-immune stromal populations (Fig. 3c).

Transcriptomic changes of the Endothelial-2 were examined within the Slide-seqV2 spatial transcriptomic platform comparing the 'tumor' and 'tumor-adjacent' contexts (Fig. 3d and S4b). Pathway enrichment analysis of Endothelial-2 was consistent with the 10X single-cell data of the tumor, showing upregulation of sprouting angiogenesis and vascular endothelial growth factor pathways (Fig. 3e, Fig. S4c).

Endothelial-2 in the tumor context also showed upregulation of cell migration and proliferation pathways (Fig. S4c). This is consistent with the dispersed organization of this subpopulation within the tumor tissue in contrast to well-organized structures of the adj-normal and healthy samples (Fig. 3f), and this was quantified by spatial auto-correlation analysis (Fig. 3g). Overall, this highlights the relevance of endothelial cells to tumor vascularization and migration which correlates with PCa progression[36].

Perivascular pericytes are another component of the vascular system. These cells exhibit mesenchymal features with multipotency[37],

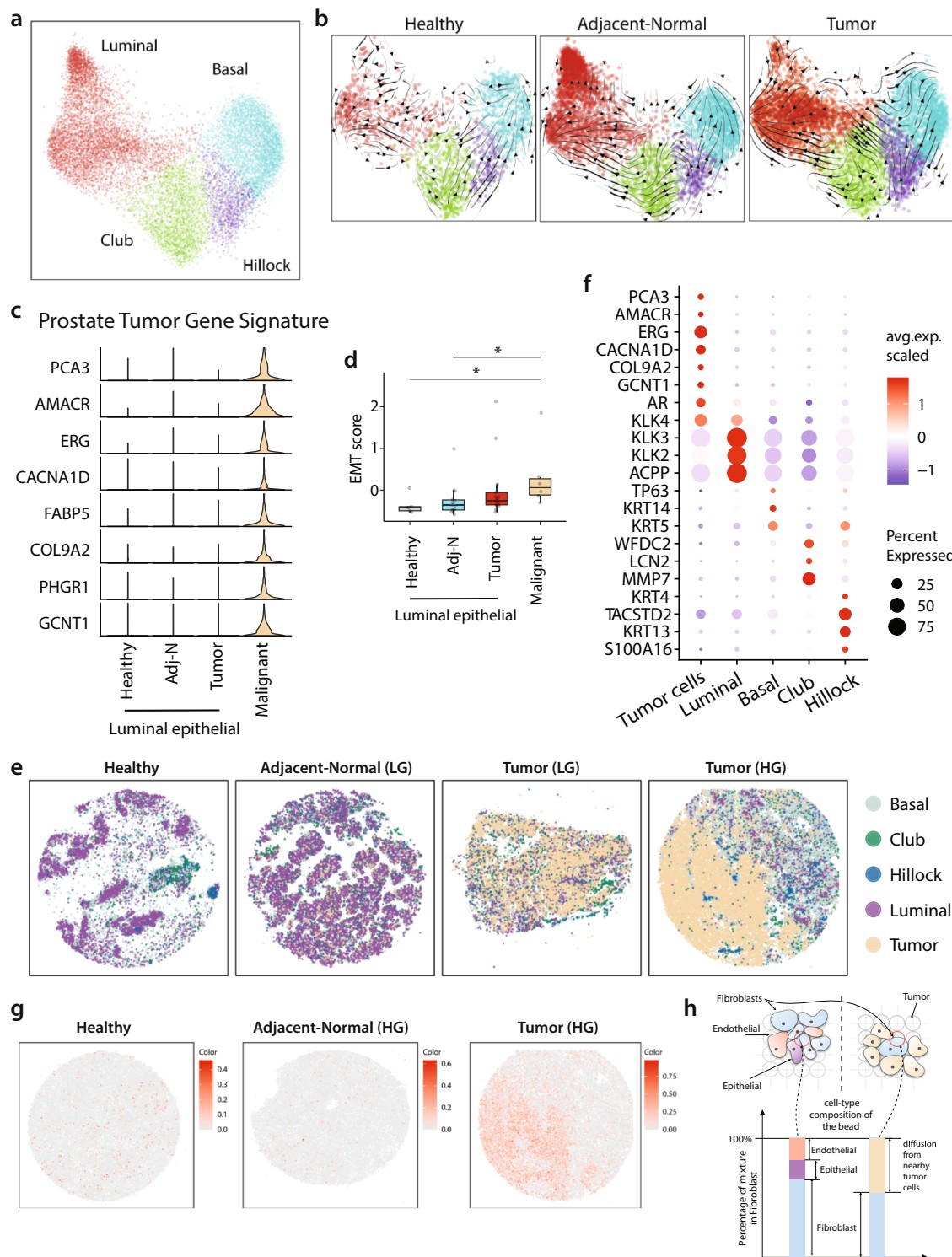

and their role in vasculature development is established while their role in cancer progression is unclear. We identified two pericyte sub-populations (Fig. 3a). The expression pattern in Pericyte-1 was enri-ched for pathways involved in extracellular structure organization and connective tissue development, while Pericyte-2 demonstrated gene signatures enriched for muscle contraction consistent with vascular smooth muscle cells (VSMCs) (Fig. 3b). In addition, there was a sig-nificant increase in the angiogenic gene signature of both pericyte subpopulations in samples collected from cancerous prostate as compared to healthy prostate (Fig. 3c). Spatially, Pericyte-1 cells were dispersed in the tumor samples when compared to healthy and adj-

normal samples (Fig. 3f and 3g). Taken together, these data suggest a role for pericytes in angiogenesis and in remodeling the tumor stroma during PCa progression.

Cancer-associated fibroblasts (CAFs) play a critical role in shaping the TME by promoting tumor proliferation and metastasis[38], enhan-cing angiogenesis[39], and mediating immunosuppression[40]. In PCa, CAFs play a causal role in cancer development at early stages, con-tributing to therapy resistance and metastatic progression[41]. Fibro-blasts gene expression patterns showed enrichment for extracellular structure organization and connective tissue development pathways (Fig. 3b). These same pathways were also identified within the Slide-

**Fig. 2 | A Prostate Tumor Gene Signature distinguishes normal and malignant luminal epithelial cells. a** Joint embedding represent the detailed annotation of epithelial subpopulations in prostate tissues. **b** RNA velocity analysis of the transitions of epithelial cells, estimated on different sample fraction. **c** Violin plot showing the expression of genes panel of "Prostate Tumor Gene Signature" in malignant cells and in the epithelial luminal cells of healthy, adj-normal, and tumor prostate samples. **d** Boxplot representing the epithelial-mesenchymal transition (EMT) score in malignant cells ($n = 6$) and the luminal epithelial cells of healthy ($n = 5$), adj-normal ($n = 14$), and tumor ($n = 17$) prostate samples. The box is bounded by the first and third quartile with a horizontal line at the median and whiskers extend to the maximum and minimum value. Significance was assessed using two-sided Wilcoxon rank-sum test (Malignant vs. adj-normal *$p = 0.02$; Malignant vs. Healthy *$p = 0.03$). **e** Spatial presentation of epithelial subpopulations in healthy ($n = 4$), adj-normal (Adj-normal LG $n = 2$) and two tumor tissues collected from low-grade (Tumor-LG $n = 2$) and high-grade (Tumor-HG $n = 2$) patients. Patinets ID from Supplementary Data 2 represented here as healthy is HP1, adj-normal of LG case is Benign04, tumor tissue of LG case is Tumor08, tumor tissue of HG case is Tumor02. **f** Dotplot representing key-marker genes expression in epithelial subpopulations in Slide-seqV2. The color represents scaled average expression of marker genes in each cell type, and the size indicates the proportion of cells expressing marker genes. **g** Spatial presentation for "Prostate Tumor Gene Signature" average expression in healthy, adjacent-normal (HG) and tumor (HG) Slide-SeqV2 pucks. **h** A schematic view of the admixture problem in the Slide-seqV2 puck. The barplot shows the cell type composition in two different contexts within the same puck. The barplot related to the tumor context contains substantial admixture from nearby tumor cells whereas the one related to tumor-adjacent context is a heterogeneous mixture of different cell types. Source data are provided as a Source Data file.

seqV2 differential gene analysis, comparing the tumor to the tumor-adjacent context (Fig. S4d). These data suggest a role for fibroblasts in inducing extracellular matrix remodeling in prostate TME, which in turn is important for tumor progression.

## Coordination between tumor cells and stromal compartment in tumor context

We utilized Slide-seqV2 spatial information to examine potential channels of communication between cells within the tumor. While the importance of cell-to-cell signaling is appreciated, it is challenging to infer which cells communicate with each other and via which channels[42]. Prediction of possible relationships is based on the expression of ligand and cognate receptor pairs and typically results in many potential interactions; additional filters are needed to distinguish functionally relevant channels. We reasoned that spatial proximity might be one such filter to identify relevant interactions.

We asked whether the corresponding ligand and receptor genes exhibited cooperative upregulation in cells positioned directly next to each. Slide-seqV2 data was used to construct a graph of physically adjacent cells, which permitted testing whether a ligand-receptor (LR) score, defined as a product of the two corresponding expression levels, was significantly higher in physically adjacent cells than would be expected from a randomized spatial arrangement (Fig. 4a, see 'Methods'). From a reference list of ~1200 ligand-receptor interactions, our analysis revealed 405 statistically significant potential communication channels (Fig. 4b, Supplementary Data 5).

With a focus on tumor-stromal communication, we investigated for communication channels when considering tumor cells as a source of ligands and stromal cells as expressing receptors (Fig. 4c). Tumor cells express vascular endothelial growth factor (*VEGFA* and *VEGFB*), which can stimulate Endothelial-2 cells through VEGF receptor, FLT1[43] and beta-1 integrin[44,45]. These channels could potentially explain the pro-angiogenic shift in the state of the tumor-associated Endothelial-2 subpopulation (Fig. 3e). We also observed potential interactions between tumor cells and fibroblasts (*COL9A2-ITGA1*) and tumor cells with Pericytes-2 cells (*COL12A1-ITGA1*) (Fig. 4c), two pathways that are both involved in extracellular matrix remodeling and cell migration[46,47].

Analysis of reverse interactions (i.e., stromal cells expressing ligand to a tumor receptor), revealed a potential interaction mediated by fibroblasts Insulin-like Growth Factor (*IGF1*) stimulating tumor cell IGF1 receptor (Fig. 4d). The IGF pathway is known to promote tumor growth and survival through suppression of apoptosis and activation of cell cycle[48]. Slide-seqV2 analysis of the *IGF1-IGF1R* interaction confirmed the co-localization of tumor cells expressing *IGF1R* and fibroblasts expressing *IGF1* (Fig. 4e).

## Immunosuppressive myeloid cells are enriched in prostate tumors

Myeloid cells support tumor progression in several cancer types, and these cells are considered one of the most clinically relevant populations to target for immune therapeutic purposes[49]. Unsupervised clustering of myeloid population revealed three monocytes, three macrophages, and 1 myeloid DC (mDC) subpopulations (Fig. 5a). Annotation was performed based on key-marker genes (Fig. 5b and S5a) and validated by monocyte and macrophage gene signatures (Supplementary Data 4, Fig. 5c, panels 1 and 2).

Monocyte subpopulations were characterized as CD16hi (CD16hi-Mo) which are known as non-classical monocytes, and tumor-inflammatory monocytes (TIMo) which had high expression of CD14, a classical monocyte marker (Fig. 5b) as well as the highest expression of an inflammatory gene signature (Supplementary Data 4, Fig. 5c, panel 3). The third subpopulation was annotated as Monocyte-Macrophage (Mo-MΦ) as it showed a gradual shift in their gene expression from genes highly expressed in monocytes (e.g., *S100A9*) to genes expressed in macrophages (e.g., *C1QA*) (Fig. S5b), suggesting a transitional cell state from monocytes toward macrophages. Both tumor and stromal cells produce chemokines involved in the myeloid differentiation process, as well as in the recruitment of monocytes to the tumor site[50]. In our dataset, we observed high expression of *CXCL12* in fibroblasts, *CCL2* in pericytes, and *CCL3,4,* and *5* in epithelial and tumor cells (Fig. S5c), suggesting a potential role of fibroblasts and pericytes in recruiting monocytes to the prostate tumor.

Patients with PCa have an immunosuppressive TME associated with the accumulation of myeloid-derived suppressor cells (MDSCs)[51,52]. In our dataset of myeloid subpopulation, TIMo cells scored highest for MDSC gene signature[53] (Supplementary Data 4, Fig. 5c, panel 5), and the gene signature was significantly higher in cells collected from cancerous prostate (tumor and adj-normal) compared to healthy prostate tissues (Fig. 5d). This suggests a role for TIMo in prostate tumor growth through immunosuppressive activity.

Several macrophage subpopulations were identified (Fig. 5a), including tumor-inflammatory macrophages (TIMΦ) with a high "Inflammatory gene signature", antigen presenting macrophages (AP MΦ) with a high "antigen processing and presentation gene signature", as well as M2-macrophages (M2-MΦ) with a high "M2-gene signature" (Fig. 5c, panels 4 and 6, Fig. S5d, Supplementary Data 4). M2-MΦ showed a gradual increase in cell abundance from healthy towards tumor fraction (Fig. 5e) and M2-macrophages have been shown to suppress anti-tumor immune response across a broad range of tumors[54]. In PCa, the high infiltration of M2-macrophages in tumor tissue has been linked to tumor recurrence[55] and metastasis[56].

Multiplex immunohistochemistry (mIHC), performed in-situ on the same tissue samples as the single-cell expression, confirmed a higher infiltration of CD68+ macrophages and of CD68+ CD163+ M2-MΦ in tumor tissues compared to their matched adj-normal tissues (Fig. 5f). Quantification of tumor infiltration by M2-MΦ was more pronounced in cases of high Gleason scores (4 + 4, 4 + 5, 5 + 5) (Fig. 5f, bottom panel). M2-MΦ express high levels of genes involved in angiogenesis such as angiogenic factor *EGFL7*[57] and in tumor

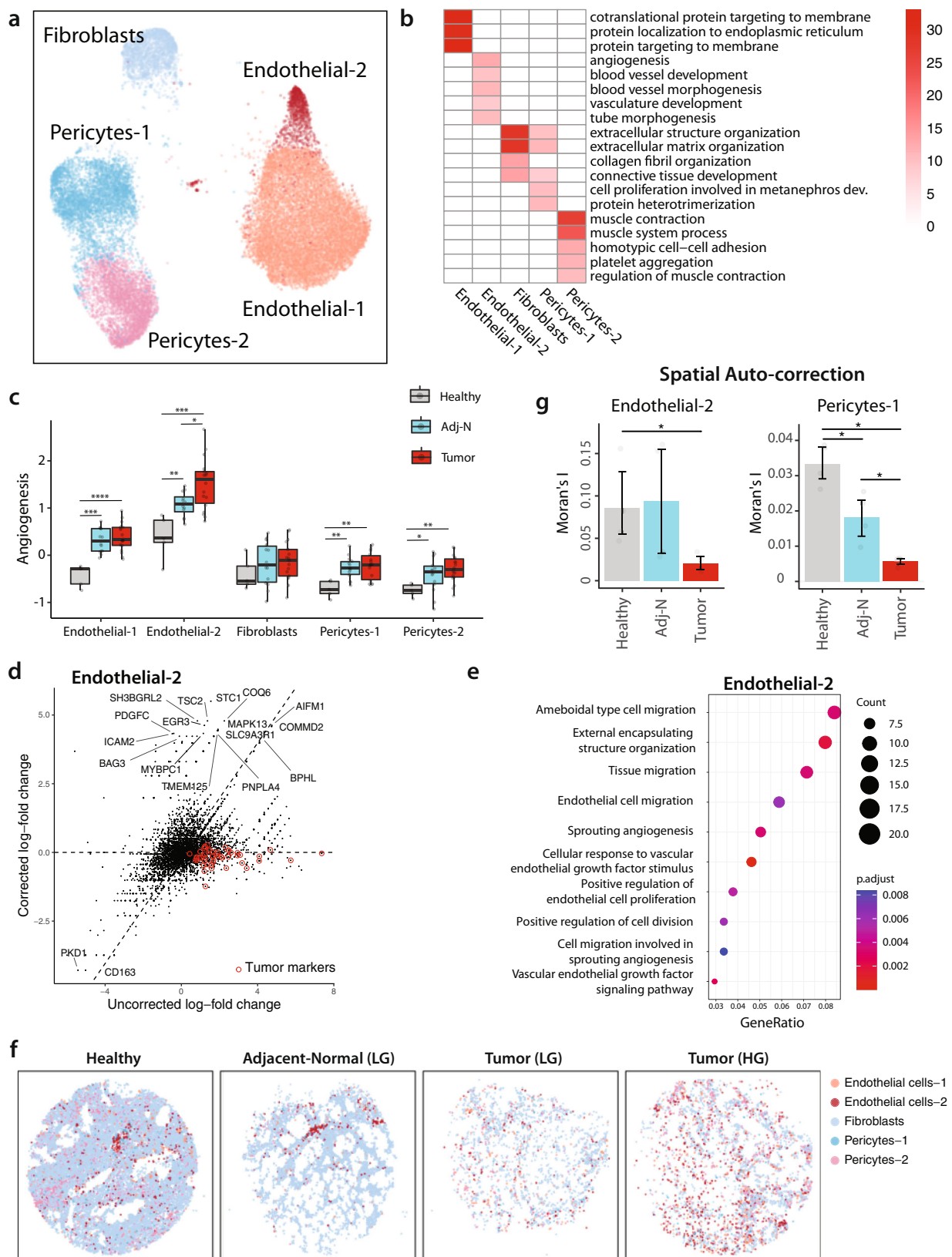

metastasis such as *LYVE1*[58] and *NRP1*[59] (Fig. S5e), suggesting a role for M2-MΦ infiltration in angiogenesis within tumors.

mDCs present tumor antigens to T-cells with a critical role in the initiation and regulation of the adaptive anti-tumor immune response[60]. We identified three mDCs subpopulations, each with high expression of either *CD1C*, *CLEC9A* or *LAMP3*. No significant changes

were observed in the cell abundance of the different mDCs subsets (Fig. S5f).

Overall, our myeloid cell analysis identified immunosuppressive subpopulations that may contribute to tumor progression, including MDSC-like monocytes (TIMo), and macrophages with an M2-gene signature.

**Fig. 3 | The prostate tumor microenvironment exhibits high endothelial angiogenic activity. a** Joint embedding represent the detailed annotation of stromal cells in prostate tissues. **b** Overview of enriched GO terms of top 200 upregulated genes for each stromal subpopulation based on single-cell data analysis. **c** Boxplot comparing the angiogenesis signature across the three different sample fractions (healthy $n = 5$; adj-normal $n = 14$; tumor $n = 18$) for each stromal subpopulation. See Supplementary Data 4 for the genes defining angiogenesis signature. Boxplots include centerline, median; box limits, upper and lower quartiles; and whiskers are highest and lowest values no >1.5× interquartile range. Statistical significance was accessed using two-sided Wilcoxon rank-sum test (*$p < 0.05$, ****$p < 0.0001$), $p$ values could be found in Supplementary Data 6. **d** The scatterplot showing the effect of linear model-based correction on Endothelial-2 cells. Red dots indicate tumor marker genes. The $x$ axis is the log-fold change of the genes without the correction, the $y$ axis is the same after the correction. The top DE genes are text-labeled. **e** Dotplot shows enriched GO terms of upregulated genes in Endothelial-2 cells in a tumor context compared to tumor-adjacent context. **f** Spatial presentation at a high-resolution level using Slide-seqV2 for the stromal subpopulations in healthy ($n = 4$), adj-normal (Adj-normal LG $n = 2$), and two tumor tissues collected from a low-grade (Tumor-LG $n = 2$) and high-grade (Tumor-HG $n = 2$) patients. Patients ID from Supplementary Data 2 represented here as healthy is HP1, adj-normal of LG case is Benign04, tumor tissue of LG case is Tumor08, tumor tissue of HG case is Tumor02. **g** Comparison of spatial autocorrelation (Moran's I) of Endothelial-2 cells and Pericytes-1 cells in healthy ($n = 4$), adj-normal ($n = 4$), and tumor samples ($n = 4$). Statistical significance was accessed using two-sided Wilcoxon rank-sum test (Endothelial cells-2 *$p = 0.03$. Pericytes-1: Tumor vs. Adj-normal *$p = 0.03$; Tumor vs. Healthy ***$p = 0.03$; Healthy vs. Adj-normal *$p = 0.03$, error bars: SEM). Source data are provided as a Source Data file. $P$ values <0.05 were considered significant: *$p < 0.05$; **$p < 0.01$; ***$p < 0.001$; ****$p < 0.0001$.

## Prostate cancer is characterized by T-cell exhaustion and immunosuppressive Treg activity

The adaptive immune system plays a pivotal role in mounting an effective, antigen-specific immune response against tumors. Unsupervised clustering of the lymphoid compartment revealed four CD4+ T-cell, three CD8+ T-cell and two NK subpopulations (Fig. 6a) as annotated by key-marker genes (Fig. 6b).

The functional state of CD8+ T-cells was assayed using a cytotoxicity gene signature ("cytotoxicity score") (Supplementary Data 4)[61,62]. CD8+ effector cells exhibited a higher cytotoxicity score compared to CTL-1 and CTL-2 (Fig. S6a). Also, we aimed to check the CTLs cytotoxicity by comparing "cold" tumors characterized with low T-cell infiltration and "hot" tumors with high degree of T-cell infiltration[63]. To this end, we collected scRNA-seq data from pancreatic ductal adenocarcinoma (PDAC)[64] as another cold tumor; and from "hot" tumors including head and neck squamous cell carcinoma (HNSCC)[65], liver hepatocelluar carcinoma (LIHC)[66] and lung cancer (lung)[3]. We first performed an integration for T-cell compartments from all datasets mentioned above where data showed the three CD8+ T-cell subsets (CTL-1, CTL-2, and CD8+ effector cells) obtained in our dataset also in the other solid tumors (Fig. S6b). Interestingly, T-cell cytotoxicity scored significantly higher in the three different CTLs in the "hot" tumors compared to "cold" tumors (Fig. 6e) that affect the efficacy of immune therapy.

In addition, we observed the low T-cell infiltration in prostate tumor in our Slide-seqV2 data (Fig. S6c and S6d). Hallmark genes denoting different cell populations were used to verify the RCTD annotation in Slide-seqV2 data (Fig. S6e).

Also, we checked for T-cell exhaustion. Both CTL-1 and CD8+ effector cells exhibited higher expression of a T-cell exhaustion gene signature[66-68] (Supplementary Data 4, Fig. 6c), and the exhaustion score was higher in the prostate tumor and adj-normal samples as compared to healthy prostate tissues (Fig. 6d).

Measurement of T-cell abundance showed a higher proportion of exhausted CTL-1 cells in tissues collected from cancerous prostate compared to healthy prostate tissues (Fig. S6f), suggesting that more CTLs in the tumor fraction became dysfunctional, and eventually acquiring an "exhausted" phenotype.

CD4+ T-cells were subdivided into naive, T-helper-1 (Th1), T-helper 17 (Th17), and T-regulatory (Treg) cells based on cell-type-specific genes[69] (Fig. 6b). CD4+ cell abundance were stable across the different sample fractions (Fig. S6f). In Tregs subpopulation, we checked for Treg activity gene expression (Treg activity score)[70,71] (Supplementary Data 4). Data showed significantly higher Treg activity score in Tregs collected from the tumor compared to adj-normal and healthy samples (Fig. 6f). Notably, genes of tumor necrosis factor receptor superfamily *TNFRSF9*, *TNFRSF18*, and *TNFRSF4* were highly and exclusively expressed in the Tregs infiltrating the tumor (Fig. S6g). These receptors bind tumor necrosis factors, pro-inflammatory cytokines involved in inflammation-

associated carcinogenesis[72] and in supporting an immunosuppressive TME.

## Coordination between myeloid and lymphoid compartments

Tregs and MDSCs represent two immunosuppressive cell populations important for cancer immune tolerance. Both populations exhibited high suppressive activity in the tumor fraction and their crosstalk has been previously reported in different cancers[73,74]. Based on this, we examined the correlation between MDSC score in TIMo and Treg activity score in Tregs both in tumor samples and their adj-normal tissues. Within the tumor fraction, the MDSC score and Treg activity score were significant correlated, with no clear separation between LG and HG Gleason patients (Fig. 7a, top). No correlation was seen in adj-normal tissues (Fig. 7a, bottom).

Based on this correlation between myeloid and T-cell suppressive populations, we checked the ligands that are highly expressed in TIMo and receptors that are upregulated in the Tregs in the tumor fraction, followed by significant potential ligand-receptor interactions between the two subpopulations. Data showed *CCL20-CCR6* as one of the significant axes of interaction between TIMo and Tregs, respectively (Fig. 7b). *CCL20* was highly expressed in TIMo and its cognate receptor *CCR6* was predominantly expressed in Tregs, Th1 and Th17 (Fig. 7c, Fig. 7d). Several studies showed an effect for *CCL20* signaling in enhancing tumor growth, invasiveness, and chemoresistance[75-77] by recruiting Tregs and/or Th17[78,79].

To examine whether CCL20-CCR6 axis is involved in the immune suppressive TME of PCa, we injected parental RM1 PCa cell line subcutaneously into C57BL/6 J wild-type (WT) mice. When tumor reached the volume of 300-400 mm³, mice were treated with CCL20-blocking antibody alone or in combination with immune check point blockade (anti-PD-1). Blocking CCL20-CCR6 interaction using CCL20-blocking antibody reduced significantly RM1 tumor growth with no combinatorial effect for anti-PD-1 (Fig. 7e), suggesting CCL20-CCR6 axis to be involved in the prostate immune suppressive TME.

Taken together, we characterized the functional status of T-cell subpopulations in prostate tumors to demonstrate exhausted CTLs along with increased Treg suppressive activity which correlated strongly with the suppressive activity of MDSC-like monocytes through CCL20-CCR6 axis.

## The prostate cancer TME is enriched in exhausted CD56^DIM NK cells

Natural killer (NK) cells are an innate lymphoid cell with cytotoxic function that can be modulated by activating and inhibitory cell-surface receptors[80]. NK cells were annotated based on key-marker gene expression (Fig. 6b)[81] and clustering revealed 4 NK subpopulations (Fig. S7a and S7b). NKT cells were characterized by high expression of T-cell marker genes *CD3D* and *CD8* and CD56^Dim NK cells by high expression of *HAVCR2*, which is expressed by terminally differentiated NK cells (Fig. S7b)[81]. CD56^bright NK cells expressed *XCL1*,

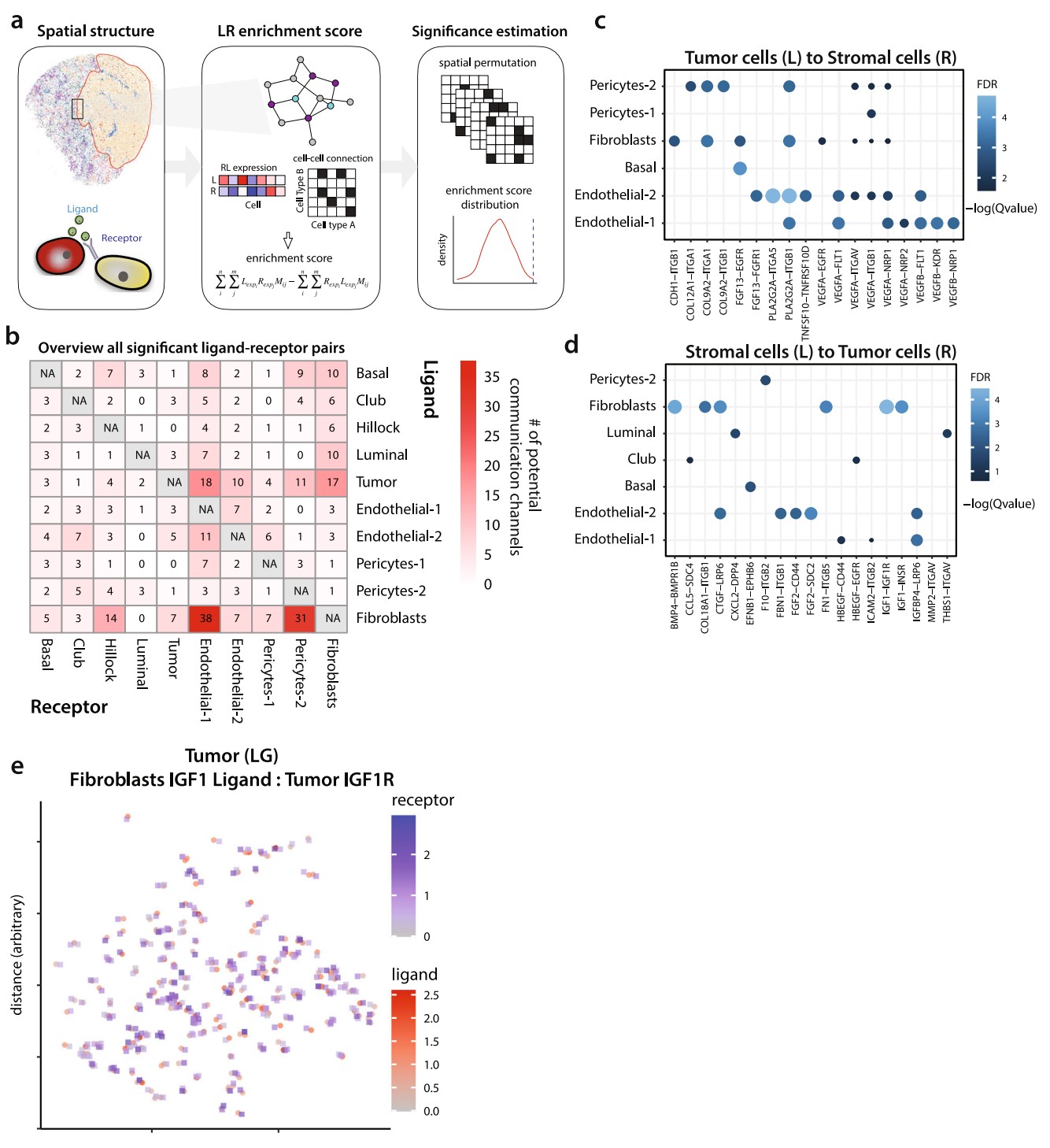

**Fig. 4 | Coordination between tumor cells and stromal compartment in tumor context. a** Schematic of ligand-receptor analysis for Slide-seqV2 data. **b** Summary of the total number of significant ligand-receptor interactions between stromal and epithelial cells. Each cell indicates potential channels of communication from ligand (row) to receptor (column). **c, d** Communication channels between tumor cells and stromal cells, communication from tumor cells (ligand) to stromal cells (receptor) (**c**), and from stromal cells (ligand) to tumor cells (receptor) (**d**). Color and size represent the significance (−log10 adjust *p* value) of ligand and receptor pairs, (e.g., Ligand IGF1 in fibroblasts and receptor IGF1Rin tumor cells). **e** Dot plot showing expression of IGF1 ligand-IGF1 receptor (IGF1R) axis in colocalized fibroblasts and tumor cells, respectively, on a low-grade (LG) tumor case. Source data are provided as a Source Data file.

*XCL2*, *GZMK*, *CD44,* and *KLRC1*[81], while the CD56[bright]-IL7R + cells separated based on specific expression of *IL7R* and the homing-receptor *SELL* (encoding CD62L)[82–84] (Fig. S7b).

The NKT and CD56[DIM] cells also showed high expression of the effector protein and cytotoxic-related genes *FGFBP2, GNLY, GZMB, GZMH*[81,85] (Fig. S7b). However, these same NK subpopulations

exhibited a higher exhaustion gene signature (Supplementary Data 4) in the tumor samples as compared to healthy tissue (Fig. S7c), suggesting impaired effector function within the prostate TME. Of the NK subpopulations, the CD56[DIM] cells scored highest for the exhaustion gene signature (Fig. S7d) and were in higher abundance in the prostate tumor as compared to the healthy prostate (Fig. S7e).

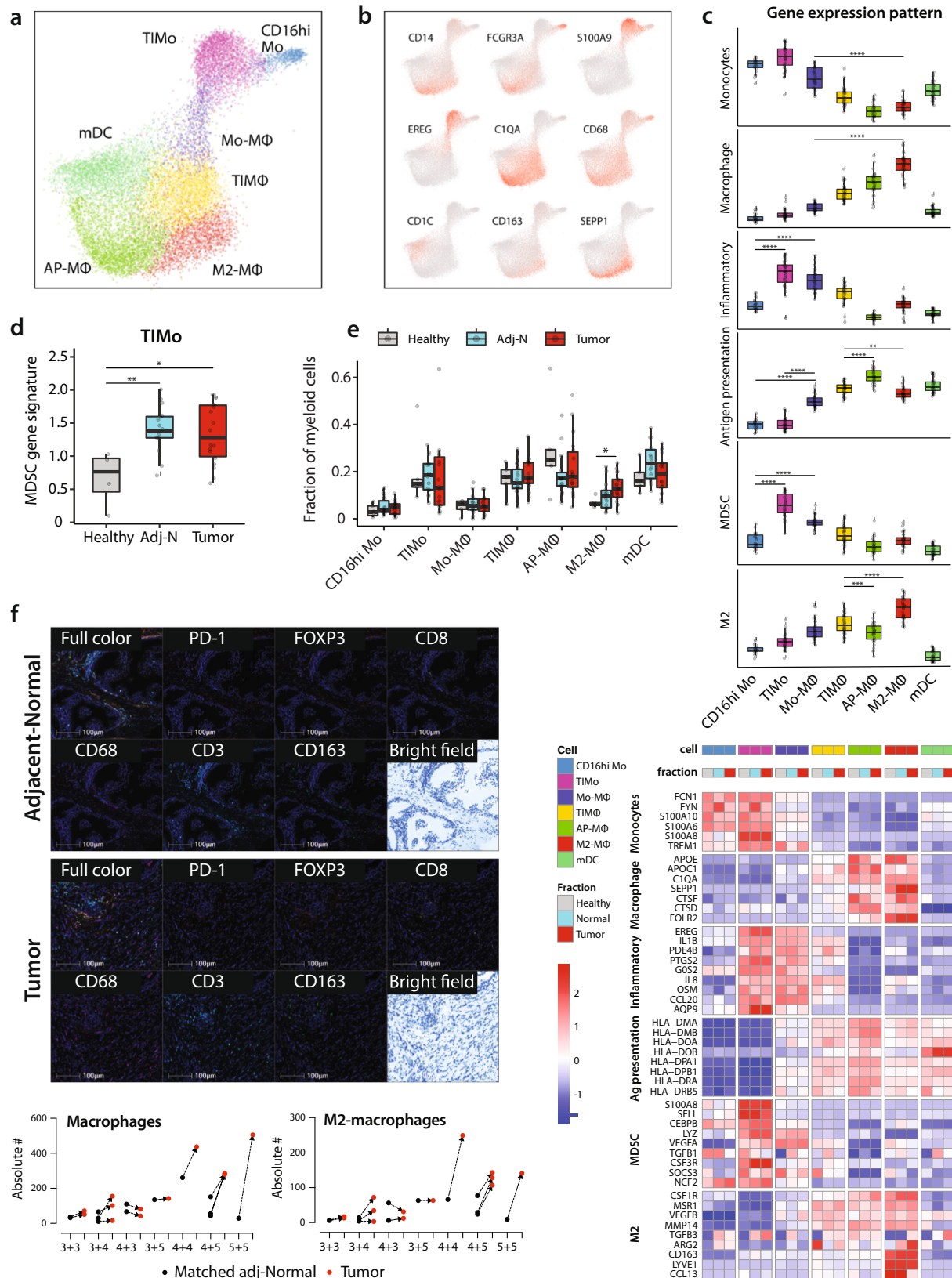

**The prostate cancer TME is characterized by activated B-cells**

B-cells are less extensively studied in cancer as compared to the myeloid and T-cell counterparts. B-cell infiltration has been described in several cancer types though their function and correlation to survival remain controversial[86]. Clustering of B-cells based on key-marker genes revealed 3 subpopulations: naive-B, active-B, and plasma cells (Fig. S7f). B-cell activity was assessed in active B-cells and plasma cells[87] (Supplementary Data 4). B-cell activity was significantly higher in cells from tumor and adj-normal tissue compared to healthy prostate (Fig. S7g), possibly due to the recognition of tumor antigens by the

**Fig. 5 | Immunosuppressive myeloid cells are enriched in prostate tumors.**
**a**, **b** Joint embedding showing the detailed annotation of the myeloid subpopulations (**a**) and the expression of select gene markers for each subpopulation (**b**).
**c** Gene expression pattern: boxplots representing the average gene expression pattern of monocyte, macrophage, inflammatory, antigen processing and presentation, MDSC gene signatures, and M2-macrophages gene signature across the different myeloid subpopulations (top). Heatmap showing the average gene expression of representative genes across the different myeloid subpopulations in healthy, adj-normal, and tumor prostate samples (bottom). See Supplementary Data 4 for the genes defining the above-mentioned signatures. **d** Boxplot comparing the average expression of MDSC gene signature in tumor-inflammatory monocytes (TIMo) across the three different samples (healthy $n = 5$, adj-normal $n = 14$, and tumor $n = 18$). **e** Boxplot representing the cell fraction of different myeloid subpopulations across the healthy ($n = 5$), tumor ($n = 18$), and their adj-normal ($n = 14$) prostate tissues. Boxplots in **c**–**e** include centerline, median; box

limits, upper and lower quartiles; and whiskers are highest and lowest values no >1.5× interquartile range. Statistical significance was accessed using two-sided Wilcoxon rank-sum test (*$p < 0.05$; **$p < 0.01$; ***$p < 0.001$; ****$p < 0.0001$), $p$ values could be found in Supplementary Data 6. **f** Top: multiplex fluorescence immuno-histochemistry (mFIHC) staining of prostate tumor tissue (bottom) and its adj-normal tissue (top) collected from a prostatectomy case of Gleason score 5 + 5. Samples are labeled with PD-1 (Clone EH33) (color Red), FOXP3 (color Orange), CD8 (color Yellow), CD68 (color Magenta), CD3 (color Cyan), CD163 (color Green), and DAPI (Blue) by using mFIHC. Bottom: quantification of absolute number of macrophages (left) and M2-macrophages (right) from mIHC data comparing tumor tissues to their matched adj-normal tissues collected from prostatectomy cases of different Gleason scores. Red circles represent the tumor samples and black circles represent their matched adj-normal samples. Source data are provided as a Source Data file.

B-cells. However, this increased activity was accompanied by a lower B-cell abundance in the tumor samples (Fig. S7h).

In our spatial characterization of immune cells, B-cells and macrophages were most abundant, with few monocytes, T-cells, and plasma cells (Fig. S6c and S6d). This low abundance did not permit a formal analysis of potential ligand/receptor interactions.

## Discussion

Localized PCa has been extensively studied using bulk transcriptomic and genomic sequencing approaches, providing insights into oncogenic drivers and recurrent molecular changes. Here, we used a high-resolution single-cell approach to characterize changes in tumor, immune, and non-immune stromal cells within the TME. These findings were complemented by spatial transcriptomic analysis where the tissue architecture and cell-to-cell relationships are preserved, allowing one to determine whether transcriptomic changes are context-dependent.

The strengths of our study include the (a) fresh nature of our patient samples, (b) matched tumor and adj-normal samples across a spectrum of Gleason scores to help overcome the inherent patient-to-patient variability, (c) rigorous collection of truly normal control prostate samples (healthy), and (d) the combined single-cell and spatial transcriptomic analysis. Indeed, this manuscript represents a highly detailed spatial transcriptomic analysis using Slide-seqV2 to characterize the prostate tumor tissue, as well as a new computational approach to detect spatial context-dependent transcriptional differences in different cell types, which are typically obscured by the admixture from neighboring cells. Such changes are likely to provide insights about the impact of microenvironment on the cell and the mechanisms through which such changes may be induced. We hope that the developed context-dependent DE method, which controls for the likely artifact of admixture from neighboring cells, will enable analysis of such context-driven changes by other investigators (Supplementary Note).

As expected, the prostate TME is complex with several subsets of myeloid cells, T-cells, NK cells, and B-cells in addition to the non-immune stromal populations of endothelial cells, fibroblasts, and pericytes. This led to some key observations.

Regarding epithelial cells, we identified distinct subsets including hillock and club cells that our RNA velocity analysis suggested a progenitor role for both subpopulations. We also used an iterative strategy to distinguish between malignant and normal epithelial cells, first relying on detection of genomic aberrations to distinguish normal and malignant luminal-type cells, and then deriving a succinct "Prostate Tumor Gene Signature", which could robustly identify tumor samples across four independent datasets (Fig. S2d).

Regarding the immune microenvironment, we obtained an immunosuppressive tumor-inflammatory monocyte with a high MDSC

gene signature. In addition, M2-macrophages were increased in abundance in the TME, a finding that was consistent across single-cell analysis and immunohistochemistry. M2-macrophages have been reported to be involved in the growth and progression of PCa and they have gained remarkable importance as therapeutic candidates for solid tumors[88]. No neutrophils were obtained within our PCa dataset. Mature neutrophils are known to have relatively low RNA content and high levels of RNases, resulting in fewer transcripts detected in Gel Bead-In EMulsions (GEMs), and less usable sequencing reads. Neutrophils are also particularly sensitive to degradation after collection or during scRNA-seq process, suggesting that we may lost the cells due to technical issues.

As for the lymphoid compartment, CTLs showed a high exhaustion signature in the tumor fraction along with higher Treg activity. Interestingly, we did not see significant T-cell differences when comparing low-grade and high-grade cases, suggesting that even the low-grade tumors had already established a highly immunosuppressive microenvironment. Even within the NK cells, the CD56$^{DIM}$ NK cells were expanded in the tumor fraction, again suggesting a functionally less cytotoxic NK cell.

Comparing the T-cell cytotoxicity of "cold" and "hot" tumors showed significantly higher cytotoxicity score in the latter (Fig. 6e), suggesting more functional T-cells that may have an effect on tumor response to immune therapy.

We hypothesized that, there is a correlation between the immunosuppressive myeloid and T-cell phenotype in our dataset as our group previously showed that immunosuppressive myeloid cells contributed to the exhausted T-cell phenotype in the setting of metastatic prostate cancer[89]. Indeed, we observed in our dataset a correlation between MDSC and Treg activity signatures in TIMo and Tregs, respectively; pointing to the role of myeloid cells in establishing a T-cell suppressive and pro-tumor microenvironment. Through computational heuristics, we identified CCL20-CCR6 as a potential ligand-receptor axis that may allow for communication between TIMo and Tregs, respectively. We showed that blocking one such signaling axis using CCL20-blocking antibody significantly reduced tumor growth in a subcutaneous model of syngeneic PCa. This axis is implicated in several inflammatory and immune-activated states, including autoimmune disease[90]. The potential for modulating the axis to reduce the activated states of immune cells has been extensively explored and led to early-stage clinical trials[91,92].

We utilized the spatial neighborhood to infer cell-to-cell interactions with high-resolution and this enabled the identification of ligand-receptor interactions in undissociated tissue section, especially between tumors cells and their stroma. Beyond the tumor-fibroblasts and tumor-endothelial cell communication that we highlighted, we hope that this analysis will prove more broadly useful for the community and point towards clinically relevant and therapeutically

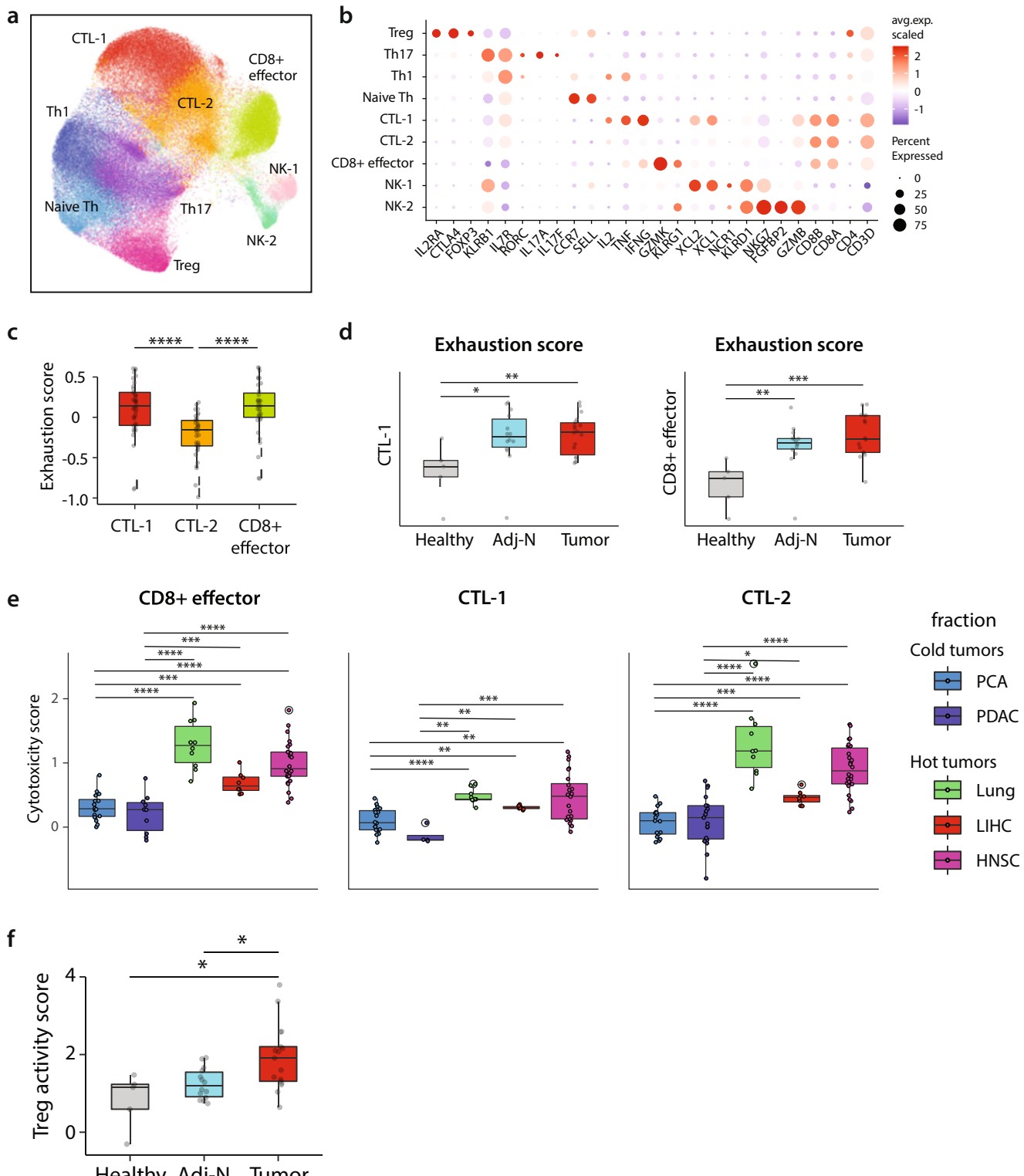

targetable interactions. This analysis also supports the complementary use of techniques that involve tissue dissociation with techniques that preserve the normal tissue architecture to home in on these cell-cell relationships.

There are several limitations that should be pointed out. First, the small sample size as total, and separating into cases of LG and HG Gleason, resulted in a lack of statistical power that masked critical elements due to heterogeneity among patients. Second, the small number of tumor cells detected based on the dissociation protocol we

used, hindered the identification of potential tumor-derived molecules that might be remodeling the immune microenvironment.

Overall, this combined dataset of single-cell and spatial transcriptomic analysis of primary prostate tumor samples and their normal controls provides a rich community resource. Biological validation of the tumor relationships with their neighboring immune and stromal cells will lead to a better understanding of prostate cancer progression and will identify new therapeutic targets for this common disease.

**Fig. 6 | Prostate cancer is characterized by T-cell exhaustion and immuno-suppressive Treg activity. a** Joint embedding showing the detailed annotation of lymphoid subpopulations. **b** Dotplot representing key-marker gene expression in lymphoid subpopulations. The color represents the scaled average expression of marker genes in each subpopulation, and the size indicates the proportion of cells expressing marker genes. **c** Boxplots represent the average expression of exhaustion score in CD8+ CTLs subpopulations (CTL-1 $n = 37$, CTL-2 $n = 36$ and CD8+ effector cells $n = 35$). Statistics are accessed with two-sided Wilcoxon rank-sum test (CTL-1 vs. CTL-2 ****$p = 3.32$E-06; CTL-2 vs. CD8 + effector ****$p = 1.61$E-06). **d** Boxplots comparing the average expression of exhaustion score in CTL-1 (left) and CD8 + effector (right) subpopulations across healthy ($n = 5$), adj-normal ($n = 14$) and tumor ($n = 18$) samples. Statistics are accessed with two-sided Wilcoxon rank-sum test (*$p < 0.05$, **$p < 0.01$, ***$p < 0.001$), $p$ values could be found in

Supplementary Data 6. **e** Boxplots represent the average expression of cytotoxicity score in CD8 + CTLs (CD8 + effector cells, CTL-1 and CTL-2) in cold tumors including prostate cancer (PCA $n = 18$) and pancreatic ductal adenocarcinoma (PDAC $n = 19$), and in hot tumors including Head and Neck squamous cell carcinoma (HNSCC $n = 26$), liver hepatocelluar carcinoma (LIHC $n = 8$) and lung cancer (lung $n = 10$). Statistical significance was accessed using two-sided Wilcoxon rank-sum test (*$p < 0.05$; **$p < 0.01$; ***$p < 0.001$; ****$p < 0.0001$), $p$ values could be found in Supplementary Data 6. **f** Boxplot represents the average expression of Treg activity gene signature in Treg subpopulation across the three different samples. Significance was assessed using two-sided Wilcoxon rank-sum test (Tumor vs. adj-normal *$p = 0.013$; Tumor vs. Healthy *$p = 0.015$). Boxplots in **c**–**f** include centerline, median; box limits, upper and lower quartiles; and whiskers are highest and lowest values no >1.5× interquartile range. Source data are provided as a Source Data file.

## Methods

### Patient materials
In accordance with the U.S. Common Rule and after Institutional Review Board (IRB) approval, all human tissues were collected at Massachusetts General Hospital (MGH, Boston, MA) and carried out with institutional review board (IRB) approval (IRB#2003P000641). Written informed consents were obtained from all participants in the study.

### Surgical approach and tumor collection
Patients with clinically localized prostate cancer were treated with minimally invasive transabdominal radical prostatectomy. The dissection of the prostate was done by antegrade approach, freeing the bladder neck, then progressing caudally to the apex and urethra. Upon freeing the prostate, it was placed in a laparoscopic specimen sac. The specimen was then immediately removed from the patient. The staff transported the tissue without delay to the pathology lab where the research staff was waiting to assure the least possible ischemic time from separation of the organ from blood supply to prepared specimen. The prostate was marked with ink, and sectioned. The prostate cancer tissue is identified by a trained genitourinary pathologist, aided with biopsy and MRI reports. The cancer is confirmed by histological examination of the immediate adjacent tissue. Cancer cell content is estimated to be 70%.

### Sample preparation
**Dissociation of tissues into single cells**. All samples were collected in Media199 supplemented with 2% (v/v) FBS. Single-cell suspensions of the tumors were obtained by cutting the tumor into small pieces (1 mm$^3$) followed by enzymatic dissociation for 45 minutes at 37 °C with shaking at 120 rpm using Collagenase I, Collagenase II, Collagenase III, Collagenase IV (all at a concentration of 1 mg/ml, Worthington Biochemical Corporation) and Dispase (2 mg/ml, Gibco) in the presence of RNase inhibitors (RNasin (Promega), RNase OUT (Invitrogen)), and DNase I (ThermoFisher). Erythrocytes were subsequently removed by ACK Lysing buffer (Quality Biological) and cells resuspended in Media199 supplemented with 2% (v/v) FBS for further analysis.

**FACS sorting.** Single cells from tumor samples were surface stained with anti-CD235-PE (Biolegend) for 30 min at 4 °C. Cells were washed twice with 2% FBS-PBS (v/v) followed by DAPI staining (1 ug/ml) (Supplementary Data 7). Flow sorting for live-nonerythroid cells (DAPI-neg/CD235-neg) was performed on a BD FACS Aria III instrument (version 8.0.3) equipped with a 100um nozzle (BD Biosciences, San Jose, CA). All flow cytometry data were analyzed using FlowJo software (FlowJo 10.8.1, Treestar, San Carlos, CA).

### Massively parallel scRNA-seq processing
Single cells were encapsulated into emulsion droplets using Chromium Controller (10X Genomics). scRNA-seq libraries were constructed using Chromium Single-Cell 3′ v2 Reagent Kit according to the manufacturer's protocol.

Briefly, the volume of the collected samples after sorting was decreased and the cells were examined and counted under a microscope with a hemocytometer. Cells then were loaded in each channel with a target output of average 4000 cells. Reverse transcription and library preparation was done on C1000 Touch Thermal cycler with 96-Deep Well Reaction Module (Bio-Rad). Amplified cDNA and final libraries were evaluated on an Agilent BioAnalyzer using a High Sensitivity DNA Kit (Agilent Technologies). Individual libraries were diluted to 4 nM and pooled for sequencing. Pools were sequenced with 75 cycle run kits (26 bp Read1, 8 bp Index1 and 55 bp Read2) on the NextSeq 500 Sequencing System (Illumina) to 70–80% saturation level.

### scRNA-seq data processing and analysis
Sequencing data were processed using 10X Cell Ranger with default parameters (version 3.0.1), aligned to GRCh37 human reference genome. In total, we obtained 187,457 cells. We removed cells with less than 600 total UMI considered as "low quality" cells. The obtained read count matrices were further analyzed with Scrublet[93] for doublets identification. Scrublet scores above 0.4 were omitted. After quality control, 179,359 cells from 39 samples were obtained. Two samples from PCA24 (PCA24-N-LG_Collagenases+Dispase, PCA24-N-LG_Rocky (Supplementary Data 2)) were only used to compare dissociation protocol and they were not added to the comparative analysis. We used Conos[13] ($k = 15$, k.self=5, matching.method = 'mNN', metric = 'angular', space = 'PCA') to integrate multiple scRNA-seq datasets together. Principal component analysis was performed on 2000 genes with the most variable expression as selected by conos. Leiden clustering was used to determine joint cell clusters across the entire dataset collection. Distances in the first 15 principal components were used to create UMAP embedding.

### Cell annotation
To annotate major cell populations in our dataset, we used sets of well-established marker genes for each of these cell populations and annotated the cells based on their high expression for the marker genes. The detailed gene list used for the cell annotations can be found in Supplementary Data 3.

For cell subpopulations assessment within each major cell population, we extracted raw count matrices and re-analyzed cell subsets separately with Conos. As an example, to annotate the myeloid subpopulation, we extracted all myeloid cells and re-analyzed the subpopulations separately with Conos (requirement of at least 40 total myeloid cells for each sample). Leiden community detection method (as implemented in Conos) was used to determine refined joint clusters, providing higher resolution than the initial analysis. UMAP embedding was estimated using embedGraph function in Conos with default parameter settings. The same analogous procedure was performed for lymphoid compartment including T-cells and NK cells as well as non-immune stromal population and epithelial cells.

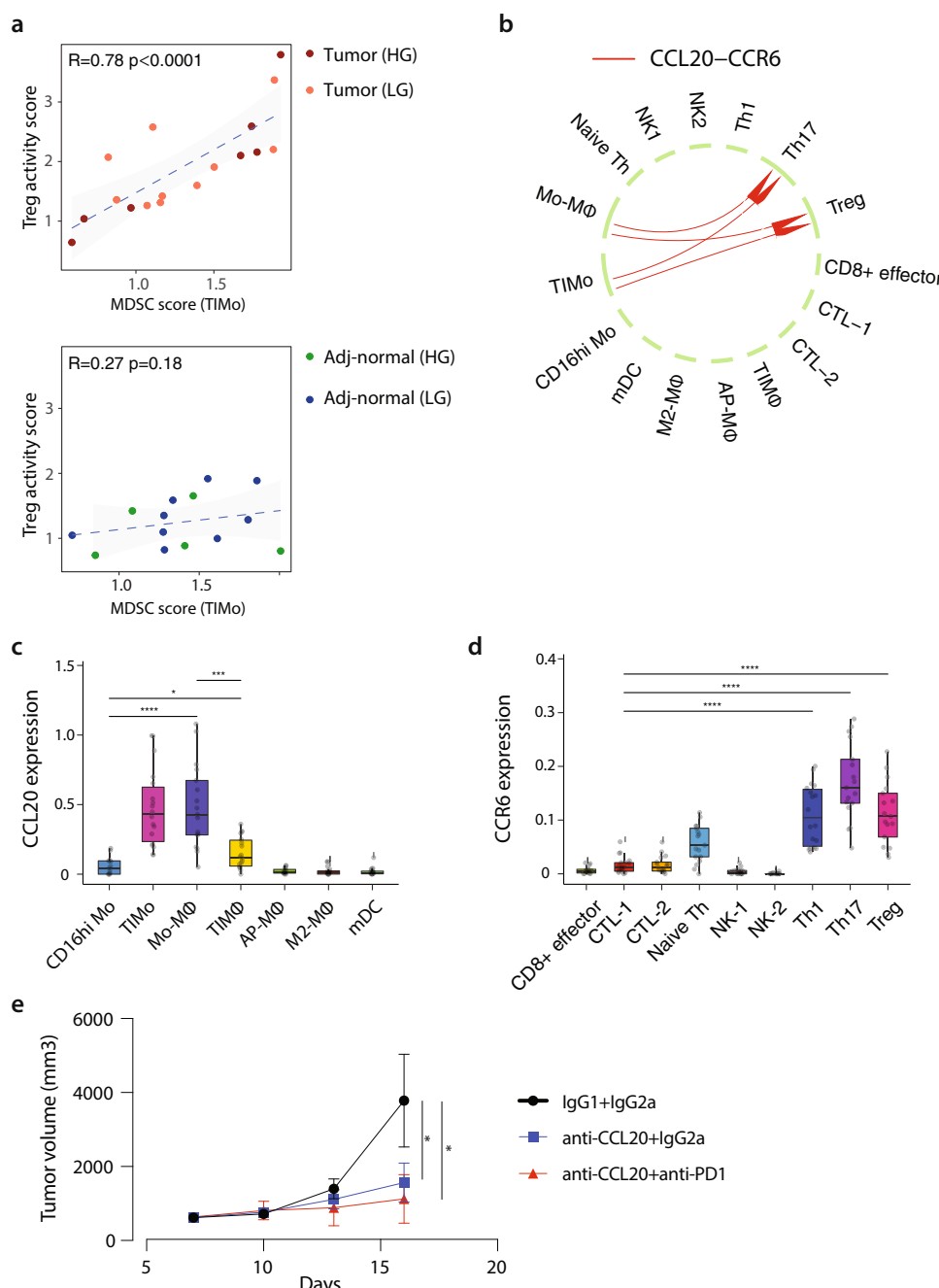

**Fig. 7 | Coordination between myeloid and lymphoid compartments. a** Scatter plot showing the correlation between Treg activity score in Tregs and MDSC score in TIMo subpopulation in tumor (top) and adj-normal prostate tissues (bottom). Each dot represents a sample. Pearson linear correlation estimate, and *p* values are shown. The error band indicates 95% confidence interval. **b** A computational approach highlighting CCL20-CCR6 interaction between myeloid and T-cell subsets. Significance of ligand-receptor pair is determined by permutation test. **c**, **d** Average expression of CCL20 (**c**) and CCR6 (**d**) is shown for different cell populations from tumor (n = 18). Statistical significance was assessed using two-sided Wilcoxon test. Boxplots in **c**, **d** include centerline, median; box limits, upper and lower quartiles; and whiskers are highest and lowest values no greater than 1.5x interquartile range. Statistical significance was accessed using Wilcoxon rank-sum test (*p < 0.05; **p < 0.01; ***p < 0.001; ****p < 0.0001), *p* values could be found in Supplementary Data 6. **e** Tumor volume of RM1 prostate tumor. Mice (5 mice/group) were injected subcutaneously with 0.25×106 RM1 cells. anti-CCL20 (200ug/kg) and/or anti-PD-1 (6 mg/kg) were injected intraperitoneally to mice every 3 days for a total of 4 times. Tumor growth was monitored by caliper measurement of the tumor volume every 3 days. Statistical significance was accessed using Wilcoxon rank-sum test (IGg1 + IGg2a vs. anti-CCL20 + anti-PD-1 *p* = 0.016; IGg1 + IGg2a vs. anti-CCL20 + IGg2a *p* = 0.029). Source data are provided as a Source Data file.

## Calculation of gene set signature scores

To assess cell states in different cell subsets and conditions, we used a gene set signature score to measure the relative difference of cell states. The signature scores were calculated as average expression values of the genes in a given set. Specifically, we first calculated signature score for each cell as an average normalized (for cell size) gene expression magnitudes, then the signature score for each sample was computed as the mean across all cells. All signature gene modules are listed in the Supplementary Data 4. The statistical significance was assessed using Wilcoxon rank-sum test.

## Differential expressed genes analysis

For DEGs analysis between cell types, Wilcoxon rank-sum test, implemented by the getDifferentialGenes() function from Conos R was used

to identify marker genes of each cell cluster. The genes were considered differentially expressed if the $p$ value determined $Z$ score was greater than 3. For DEGs analysis between sample fractions (for example Tumor Treg vs. adj-Normal Treg), getPerCellTypeDE() function in Conos R package was utilized with default settings. DESeq2[94] was applied to "mini-bulk" (or meta-cell) RNA-seq measurements by combining all molecules measured for each gene in each subpopulation in each sample. A minimal number of 10 cells (of the selected cell type) were required for a sample to be included in the comparison.

## Identification of malignant epithelial cells
To identify the malignant cells from non-malignant epithelial cells, we used interCNV[21,95] for inferring large-scale chromosomal copy number variations. We ran inferCNV on different epithelial subpopulations using the same cell type from healthy tissues as the reference "normal" cells. Only epithelial luminal cells showed clear copy number aberration. To identify malignant cells, we examined hierarchical clustering of CNV profiles obtained from inferCNV and selected malignant cells with deletion in chr8, chr12 and chr16. The set of tumor cells was then expanded based on the "prostate cancer signatures" (see next section). In total, 1237 malignant cells were classified.

## Generation of the "Prostate Tumor Gene Signature"
To generate a gene expression signature that is clinically relevant, we compared the gene expression profiles between malignant cells and non-malignant luminal cells in tumor fraction. Only the upregulated genes with $Z$ score >3 were selected and taken into subsequent analysis. We next screened each of the DEGs based on their expression in healthy prostate tissue, requiring each gene to be expressed in less than 5% cells of all epithelial cells. In total, we identified 8 significant DEGs that met the above criteria. The average expression of these curated DE genes is regarded as the diagnosis signature score, later used on multiple bulk RNA-seq data to quantify the predictive accuracy of such signature. ROC analysis showed a strong prostate cancer predictive ability with an Area Under the Curve (AUC) score of 0.956 for GSE21034[23], 0.93 for GSE97284[24], 0.937 for TCGA[25] (https://www.cbioportal.org/study/clinicalData?id=prad_tcga) and 0.94 for GSE70770[26] in four independent prostate cancer cohorts. We then applied the "Prostate Tumor Gene Signature" to all tumor luminal cells. The tumor signature scores were calculated as average expression values of the 8 significant DEGs (Prostate Tumor Gene Signature) for each cell, with additional 201 tumor cells identified based on the threshold 0.1.

## Clustering of malignant cells
To examine the tumor cells heterogeneity, we extracted tumor cells and re-ran data integration using Conos requiring at least 40 cells for each sample. In short, each individual dataset was first normalized and projected to low-dimension space using ICA. Different samples were then aligned together using Conos. UMAP embedding was estimated using default parameter settings. Leiden clustering (conos::findCommunities) was used to determine joint cell clusters across the entire dataset collection. We identified three tumor cell subclusters (C1-C3).

## RNA velocity-based cell fate tracing
To perform the RNA velocity analysis, the spliced reads and unspliced reads were recounted by the velocyto python package[96] based on previous aligned bam files of scRNA-seq data. The calculation of RNA velocity on the UMAP embedding were done by following the scVelo[19] pipeline on both individual sample group as well as the merged dataset.

## Multiplex immunohistochemistry analysis
We used multiplex immunohistochemistry (mIHC) panel to evaluate a set of unselected radical prostatectomy cases, spanning all grade groups. A seven-plex Fluorescence Immunohistochemistry assay was performed on 4 μm FFPE sections, using Leica Bond Rx autostainer. A six antibodies panel consisted of CD3 (Rabbit polyclonal, Dako), CD8 (C8/144B, Mouse monoclonal, Dako), PD-1(EH33, Mouse monoclonal, Cell Signaling), FOXP3 (D2W8E, Rabbit monoclonal, Cell Signaling), CD68 (PG-M1, Mouse monoclonal, Dako), CD163 (10D6, Mouse monoclonal, Leica Biosystem), along with DAPI counterstaining. Briefly the staining consists of sequential tyramine signal amplified immunofluorescence labels for each target, and a DAPI counterstain. Each labeling cycle consists of application of a primary antibody, a secondary antibody conjugated to horse radish peroxidase (HRP), and an opal fluorophore (Opal 690, Opal 570, Opal 540, Opal 620, Opal 650 and Opal 520, Akoya Biosciences), respectively. The stained slides were scanned on a Perkin Elmer Vectra 3 imaging system (Akoya Biosciences) and analyzed using Halo Image Analysis platform (Indica Labs). Each single stained control slide is imaged with the established exposure time for creating the spectral library. We ran an algorithm learning tool utilizing the Halo image software training for the gland and stroma regions, and subsequently completed cell segmentation. The thresholds for the antibodies were set respectively, based on the staining intensity, by cross reviewing more than 20 images. Cells with the intensity above the setting threshold were defined as positive. Regions of interest included both immune-cell-rich and non-rich areas and included both tumor and benign areas.

## Comparative analysis with public scRNA-seq datasets of different cancer types
To compare the profile of lymphoid compartment of our prostate cancer dataset with other cancer types, we collected single-cell data from pancreatic ductal adenocarcinoma (PDAC)[64], lung cancer (LUSC)[3], liver hepatocellular carcinoma (LIHC)[66], head and neck squamous Cell Carcinoma (HNSC)[65], then performed a joint alignment for lymphocyte compartment. Prostate cancer cell annotations were propagated to other cancer types using propagateLabels() function in Conos followed by comparative analysis for T-cell cytotoxicity using "cytotoxicity score".

## CCL20-blocking antibody treatment
RM1 prostate cancer cells ($0.25 \times 10^6$ cells) were injected subcutaneously to C57BL6/J male mice (#000664) from The Jackson Laboratory. When tumor reached the volume of 300–400 mm³, mice received an intraperitoneal injection of 45 μg anti-CCL20-blocking antibody (R&D Systems, clone 114908) or rat IgG isotype control antibody (R&D Systems, clone 43414), 150 μg of anti-mouse PD-1 (BioXcell, clone RMP1-14) or rat IgG2a isotype control (BioXcell, clone 2A3). The mice were treated every 3 days for a total of 3 times. Tumor growth was monitored by caliper measurement of the tumor volume and data were analyzed using Prism Software (Version 9.5.0 (525)). Statistical analyses were performed as indicated and $p$ values of ≤0.05 considered significant.

RM1 cells were maintained in DMEM (Corning, 15-013-CV) complemented with 10% FBS (GIBCO by Life Technologies, A31605-01) and 1% Penicillin-Streptomycin (GIBCO by Life Technologies, 15140-122). All mice were maintained in pathogen-free conditions and all procedures were approved by the institutional Animal Care and Use Committee of Massachusetts General Hospital. Statistical analyses were performed as indicated and $P$ values of ≤0.05 considered significant.

## Slide-seqV2 processing and sequencing
Slide-seq arrays were prepared and spatial bead barcodes sequenced following Slide-seqV2[11] protocol, using arrays (named pucks) created with custom synthesized barcoded beads (5'-TTT_PC_GCCGGTAAT ACGACTCACTATAGGGCTACACGACGCTCTTCCGATCTJJJJJJJTCTTCA GCGTTCCCGAGAJJJJJJJNNNNNNNVVT30-3') with a photocleavable linker, a bead barcode sequence (J, 14 bp), a UMI sequence

(NNNNNNNNVV, 9 bp), and a poly dT tail. Slide-seqV2 technique was applied on 4 different patients- a total of 12 samples: 2 healthy tissues collected from cystoprostatectomy surgeries (patients with bladder cancer), one tumor tissue of Gleason 4 + 3 and one tumor tissue of Gleason 5 + 4 collected from prostatectomy surgeries along with their adjacent-normal tissues (Supplementary Data 2). Two samples from each tissue were collected.

OCT-embedded frozen tissue samples were warmed to −20 °C in a cryostat (Leica CM3050S) and serially sectioned at a 10 µm thickness (2–3 Slide-seq array replicates per sample), with consecutive sections used for hematoxylin and eosin staining. Each tissue section was affixed to an array and moved into a 1.5 ml eppendorf tube for downstream processing. Briefly, the samples library was prepared as the following:

**RNA hybridization.** Pucks in 1.5-ml tubes were immersed in 200 µl of hybridization buffer (6µ SSC with 2 U/µl Lucigen NxGen RNase inhibitor) for 30 min at room temperature to allow for binding of the RNA to the oligonucleotides on the beads.

**First-strand synthesis.** First-strand synthesis was performed by incubating the pucks in RT solution (115 µl water, 40 µl Maxima 5× RT buffer (Thermo Fisher, EP0751), 20 µl of 10 mM dNTPs (NEB, N0477L), 5 µl RNase inhibitor (Lucigen, 30281), 10 µl of 50 µM template switch oligonucleotide (Qiagen, 339414YCO0076714) and 10 µl Maxima H Minus reverse transcriptase (Thermo Fisher, EP0751) for 1.5 h at 52 °C.

**Tissue digestion.** 200 µl of 2× tissue digestion buffer (200 mM Tris-Cl pH 8, 400 mM NaCl, 4% SDS, 10 mM EDTA and 32 U/ml proteinase K (NEB, P8107S)) was then added directly to the RT solution, and the mixture was incubated at 37 °C for 30 min.

**Second-strand synthesis.** The solution was then pipetted up and down vigorously to remove beads from the surface, and the glass substrate was removed from the tube and discarded. 200 µl of wash buffer (10 mM Tris pH 8.0, 1 mM EDTA and 0.01% Tween-20) was then added to the 400 µl of tissue clearing and RT solution mix, and the tube was centrifuged for 3 min at 3000 × g. The beads were washed in 200 µl of wash buffer for a total of three times then resuspended in 200 µl of ExoI mix (170 µl water, 20 µl ExoI buffer, and 10 µl ExoI (NEB, M0568)) and incubated at 37 °C for 50 min.

After ExoI treatment, the beads were centrifuged for 3 min at 3000 × g and washed in 200 µl of wash buffer for a total of three times then resuspended in 200 µl of 0.1 N NaOH and incubated for 5 min at room temperature. To quench the reaction, 200 µl of wash buffer was added and beads were centrifuged for 3 min at 3000 × g. This was repeated for a total of three times. Second-strand synthesis was then performed on the beads by incubating the pellet in 200 µl of second-strand synthesis mix (133 µl water, 40 µl Maxima 5× RT buffer, 20 µl of 10 mM dNTPs, 2 µl of 1 mM dN-SMRT oligonucleotide and 5 µl Klenow enzyme (NEB, M0210)) at 37 °C for 1 h. After second-strand synthesis, beads were washed in 200 µl of wash buffer a total of three times.

**Library amplification.** Beads were resuspended in 200 µl water and transferred into a PCR strip tube, pelleted in a minifuge, then resuspended in library PCR mix. PCR was performed following the program of 1 cycle of 98 °C for 2 min, 4 cycles of 98 °C for 20 s, 65 °C for 45 s, 72 °C for 3 min, 11 cycles of 98 °C for 20 s, 67 °C for 20 s, 72 °C for 3 min, and 1 cycle of 72 °C for 5 min. The PCR was performed in a final volume of 200 µl of PCR mix, divided into 4 PCR tubes.

**PCR cleanup and Nextera tagmentation.** The PCR product was then purified by adding 30 µl of Ampure XP (Beckman Coulter A63880) beads to 50 µl of PCR product. The samples were cleaned according to manufacturer's instructions and resuspended into 50 µl water and

the cleanup was repeated resuspending in a final concentration of 10 µl. 1 µl of the library was quantified on an Agilent Bioanalyzer High sensitivity DNA chip (Agilent 5067-4626). Then, 600 pg of PCR product was taken and prepared into Illumina sequencing libraries through tagmentation with Nextera XT kit (Illumina FC-131-1096). Tagmentation was performed according to manufacturer's instructions and the library was amplified with primers Truseq5 and N700 series barcoded index primers. PCR was performed following the program of 72 °C for 3 minutes, 95 °C for 30 seconds, 12 cycles of 95 °C for 10 seconds, 55 °C for 30 seconds, 72 °C for 30 seconds then 72 °C for 5 minutes and hold at 10 °C. Samples were cleaned with AMPURE XP (Beckman Coulter A63880) beads in accordance with manufacturer's instructions at a 0.6× bead/sample ratio (30 µl of beads to 50 µl of sample) and resuspended in 10 µl water. Library quantification was performed using the Bioanalyzer. Libraries were sequenced using the following read structure on a NovaSeq (S2; Illumina): Read1: 42 bp; Read2: 41 bp; Index1: 8 bp, and sequences were processed using the pipeline available at https://github.com/MacoskoLab/slideseq-tools.

We used the Broad Institute pipeline (from https://github.com/MacoskoLab/slideseq-tools.git) to generate the count matrices and the bead locations from the raw BCL files.

### Slide-seq data processing and cell-type annotation

Sequencing data were processed using Slideseq-tools pipeline (https://github.com/MacoskoLab/slideseq-tools). First the raw sequence data is aligned to human genome reference version hg38 to obtain count matrixes and beads spatial coordinates. We used recently published RCTD[15] to annotate spatial barcoded beads. Specifically, we sampled down 10X scRNA-seq data to 1,000 cells per cell type and transfer the 10X data into the RCTD object as reference. Slide-seqV2 data were filtered using default RCTD setting, requiring at least 100 UMI per cell. To annotate Slide-seq beads. We first annotated the major cell clusters (T-cells, B-cells, stromal cells, epithelial cells and myeloid cells) with corresponding 10X reference in major cell annotation, then each of the major cell cluster was extracted for cell sub-cluster annotation. We only keep the spatial beads that are predicted as "singlet" or "doublet-certain" categories.

### Spatial autocorrelation analysis

To measure how the cells are spatially distributed across the puck, we measured the spatial autocorrelation metric and evaluated clustering centrality pattern for each cell type. We applied "compute auto-correlations" function from hotspot package[97], and calculated the Moran's I score to capture the overall spatial sparsity of cell-type-specific spatial distribution. The positive value of Moran's I score indicates the centralized clustering whereas the lower score signifies the lack of centralization. Finally, Wilcoxon signed-rank test was used to access Moran's I differences across healthy, adj-normal and tumor conditions.

### Estimate spatially differential expressed genes

To obtain the differentially expressed genes across different regions within a puck, we used a custom pre-processing phase. We first identified specific regions within the tumor puck by segmenting out the tumor proliferated region as "tumor context" and the non-proliferated region within the puck as "tumor-adjacent context". The context-specific cell level expressions were then summarized to the cell-type level pseudo-bulk profiles. We used a constrained linear regression model to correct for the linear ad-mixture effects in the Slide-seqV2 measurement given a target cell-type. Finally, the corrected pseudo-bulk profiles were used to carry out differential expression test with the standard edgeR package functions[98]. For a detailed overview of the differential expression pipeline please refer to the Supplementary Note.

## Identification of significant ligand-receptor pairs

Following the widely used protocol of delineating the significant ligand-receptor (LR) identification, we used the already LR pairs downloaded from CellPhoneDB (v1.1.0)[99] as a background. In 10X data, the significant LR was discovered by examining ligand and receptor expression in 'sender cells' and 'receiver cells'. Specifically, we first calculated the gene expression ratio scores for each cell type, considering the genes that are at least expressed in 10% of cells within that cell type. To obtain the signal strength of a LR pair in two corresponding cell types, we relied on the join expression distribution of the associated genes. Specifically, we computed the LR pair score given a cell type A and cell type B as the product of average expression of the ligand from cell type A and receptor from cell type B. We observed that such product might lead to an inflation of LR pairs that are in actual not present in the environment. To select for the statistically significant interactions, we further randomly shuffled the cluster labels of all cell types and re-calculated LR pair score across 1,000 permutations. This background is used as null distribution to evaluate the P value for the target LR pair interactions.

To access ligand-receptor interactions in Slide-seqV2 data, we combined information from the spatial structure of the cell-types in conjunction with the ligand-receptor expression. The assumed that spatially inferred ligand-receptor pairs should be co-expressed in adjacent cells. To test for such interactions, we first build a k-nearest neighbor graph (kNN, $k = 10$) based on the spatial coordinates of the corresponding beads, then for any pair of cell types, we defined a LR pair score to filter significant LR pairs by calculating the aggregated expression product of ligand and receptor in adjacent neighborhood cells obtained from kNN graph.

Formally, LR pair score for cell types A and B respectively was defined as:

$$S = \sum_i^n \sum_j^m Lexp_i * Rexp_j * M_{ij} - \sum_i^n \sum_j^m Rexp_i * Lexp_j * M_{ij} \quad (1)$$

Here n represents the number of cells for the potential "sender" cell type A, m represents the number of the "receiver" cells of the cell type B. $L\exp_i$ represents Ligand L expression in a cell of the cell type $A_i$. $R\exp_j$ represents Receptor R expression in a cell of a cell type $B_j$. $M_{ij}$ is spatial graph edge matrix, specifying whether the two cells are spatially proximal to each other. To avoid potential bias from admixture noise, such as the ligand expression signal from "receiver" cell type B and the receptor expression signal from "sender" cell type A, the score also incorporates a reverse expression that swaps the ligand and the receptor. $R\exp_i$ represents Receptor R expression in a cell of the cell type $A_i$. $L\exp_j$ represents Ligand L expression in a cell of the cell type $B_j$. To evaluate if the LR pair score S is statistically significant, we estimated a background distribution by shuffling cell labels in expression matrix (shuffling happens in 2000 rounds). In each round, a permitted score S was calculated using the same formula. P values were then estimated as an empirical tail probability of observed LR pair score S given the background distribution. The p values for all LR pairs corresponding to the cell-types were subsequently adjusted for multiple hypothesis testing. In total, 405 significant potential interaction were detected (Supplementary Data 5).

## Reporting summary

Further information on research design is available in the Nature Portfolio Reporting Summary linked to this article.

## Data availability

Raw single-cell RNA sequencing data and processed data can be accessed from the NCBI Gene Expression Omnibus database GSE181294 (https://www.ncbi.nlm.nih.gov/geo/query/acc.cgi?acc= GSE181294). GRCh38 human reference genome was download from 10X genomics (https://support.10xgenomics.com/single-cell-gene-expression/software/downloads/). For the joint alignment analysis with public scRNA-seq data. We downloaded raw count matrix for PDAC (https://ngdc.cncb.ac.cn/bioproject/browse/PRJCA001063), LIHC (https://www.ncbi.nlm.nih.gov/geo/query/acc.cgi?acc=GSE140228), LUSC (https://www.ebi.ac.uk/biostudies/arrayexpress/studies/E-MTAB-6149), HNSC (https://www.ncbi.nlm.nih.gov/geo/query/acc.cgi?acc= GSE139324). Prostate cancer bulk RNA-seq and microarray data are download from the NCBI Gene Expression Omnibus database: GSE21034, GSE97284, TCGA (https://www.cbioportal.org/study/ clinicalData?id=prad_tcga), GSE70770. All other relevant data supporting the key findings of this study are available within the article and its Supplementary Information files. Source data are provided with this paper.

## Code availability

Custom code that was used in this study can be found on github at https://github.com/shenglinmei/ProstateCancerAnalysis and Zenodo at https://zenodo.org/record/7526696#.Y78bfOzMJTZ. In addition, we created an interactive web atlas to disseminate the data. Raw count matrixes and cell annotations are also available at the github page.

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

## Acknowledgements

We are particularly indebted to our patients and their clinical care teams. We gratefully acknowledge support from Bill & Cheryl Swanson, Gunther & Maggie Buerman, the Cygnus Montanus Foundation founded by the Svanberg family, the Sullivan Family Foundation, Frank & Hanne Castle, and Robert Higginbotham. We acknowledge funding from NIH CA193481 and DK103074 to D.T.S. and P.V.K., National Cancer Institute CA 163191 to D.T.S., NIH R01HL131768 to P.V.K., European Research Council Synergy ('KILL OR DIFFERENTIATE', 856529, ERC-2019-SyG) to P.V.K., Dana-Farber / Harvard Cancer Center Nodal Award (CCSG grant P30CA006516), the Harvard Ludwig Cancer Center, the Harvard Stem Cell Institute and the Gerald and Darlene Jordan Professor of Medicine Chair to D.T.S. N.B. was funded by the Swedish Cancer Society and Swedish Childhood Cancer Fund. Y.K. was supported by a grant from the STARR cancer consortium. K.S. was supported by a grant from Urology Care Foundation Research Scholar Award and Prostate Cancer Foundation Young Investigator Award. Olga Kharchenko designed the medical illustration in Fig. 1a. Patient samples were sorted at the HSCI/CRM flow cytometry core facility at MGH.

## Author contributions

T.H., N.B., D.M.D., D.B.S., and P.J.S. conceived the study. P.J.S. coordinated the multi-disciplinary teams and the IRB-approved protocol. T.H., S.M., N.B., P.J.S., D.B.S., and P.V.K. directed the study. Sample collection methodology and surgeries were performed by D.M.D. D.Z. and M.W. provided healthy prostate tissues from cystoprostatectomy cases. D.M.D. provided the prostate tissues from prostatectomy cases.

Pathological analysis was performed by C.W., S.W., and A.S. Human samples were collected and isolated, and libraries were prepared by T.H., N.B., and Y.K. In vivo experiments were performed, and data were collected and analyzed by T.H. Slide-seq arrays, and library preparation was performed by E.M. in the labs of F.C. and E.Z.M at the broad. S.M., H.S., and P.V.K. performed the computational analysis. T.H., S.M., H.S., B.V., K.S., N.B., D.B.S., and P.V.K. interpreted the data. T.H., S.M., H.S., D.B.S., and P.V.K. wrote the manuscript. N.B., P.J.S., P.V.K., and D.B.S. jointly supervised this work. All authors read, edited, and approved the manuscript.

## Competing interests

A.O.S. own shares in TScan Therapeutics and BioNTech. P.V.K. serves on the Scientific Advisory Board to Celsius Therapeutics Inc. and Biomage Inc. P.V.K. is an employee of Altos Labs. D.T.S. is a founder, director, and stockholder of Magenta Therapeutics, Clear Creek Bio, and LifeVaultBio. He is a director and stockholder of Agios Pharmaceuticals and Editas Medicines and a founder and stockholder of Fate Therapeutics and Geruda Therapeutics. He is a consultant for FOG Pharma, Inzen Therapeutics, ResoluteBio, and VCanBio and receives sponsored research support on an unrelated project from Sumitomo Dianippon. D.B.S. is a founder, consultant, and shareholder for Clear Creek Bio. K.S. is a recipient of sponsored research funding from Convergent Genomics. F.C. and E.Z.M. are consultants for Atlas Bio, inc. The remaining authors declare no competing interests.

## Additional information

[1]Center for Regenerative Medicine, Massachusetts General Hospital, Boston, MA, USA. [2]Harvard Stem Cell Institute, Cambridge, MA, USA. [3]Department of Stem Cell and Regenerative Biology, Harvard University, Cambridge, MA, USA. [4]Department of Biomedical Informatics, Harvard Medical School, Boston, MA, USA. [5]Department of Pathology, Massachusetts General Hospital, Harvard Medical School, Boston, MA, USA. [6]Department of Urology, Massachusetts General Hospital, Harvard Medical School, Boston, MA, USA. [7]Childhood Cancer Research Unit, Department of Women's and Children's Health, Karolinska Institutet, Stockholm, Sweden. [8]Broad Institute of Harvard and MIT, Cambridge, MA, USA. [9]Department of Psychiatry, Massachusetts General Hospital, Boston, MA, USA. [10]Massachusetts General Hospital Cancer Center, Harvard Medical School, Boston, MA, USA. [11]Present address: Altos Labs, San Diego, CA, USA. [12]These authors contributed equally: Taghreed Hirz, Shenglin Mei, Ninib Baryawno, Philip J. Saylor, Peter V. Kharchenko, David B. Sykes. ✉e-mail: THIRZ@mgh.harvard.edu; smei8@mgh.harvard.edu; DBSYKES@mgh.harvard.edu

