## [Peer Review File · Nature Communications]

Dissecting the immune suppressive human prostate tumor microenvironment via integrated single-cell and spatial transcriptomic analysesREVIEWER COMMENTS

Reviewer #1 (Remarks to the Author):

In this paper, the authors investigated the TME heterogeneity of localized prostate cancer with integrated single cell sequencing and spatial transcriptome technology. Their analysis revealed an immunosuppressive microenvironment associated with suppressive myeloid populations and T cell exhaustion, along with high stromal angiogenic activity. These findings could help us understand the heterogeneity of prostate cancer. However, we still have several concerns:

Major concern:

In general, this work seems descriptive but lacks novel findings. Prostate cancer is a well-known "cold" tumor, which contains few immune cells. Although the authors find many immunosuppressive cell types in the TME, most of these findings have already been reported in studies on other tumor models. These results still could not explain why prostate cancer is "cold". And it also could not explain the difference between prostate cancer and other solid tumors.

Minor concern:

1. Figure 1D, color of this bar plot is really confusing. Annotation should be added.
2. Stroma cells annotated as pericyte by the authors contain two different subpopulations. It may be able to be divided into pericyte and SMC with conventional marker, such as MYH11. And as shown in Figure 1B, it seems like that one population was enriched in tumor tissues. However, it's difficult to know which one was that in current result.
3. Line 283, the expression should be modified. "Immunosuppressive myeloid cells are enriched in prostate tumors" may be better.
4. The hyperlink in Data Availability only contains 3' single cell sequencing data, while slide-seq V2 data was not uploaded.
5. Page 50. It's difficult to understand the meaning of the figure legend of Figure S7. Was there a writing mistake?

Reviewer #2 (Remarks to the Author):

Major comments:

Their introduction starts with the important contrast between indolent versus aggressive prostate cancer. In their study, they segregated tumors into high grade (HG) versus low grade (LG). While HG vs. LG and aggressive vs. indolent are highly related, they are not the same thing [PMID: 24027026]. It would be very useful if their data can shed new insights into differences between indolent versus aggressive disease. Rather, many of their findings could not differentiate high-grade versus low grade tumors, let alone indolent versus aggressive.

Another important point in their introduction is "The prostate TME typically contains few immune cells..." as compared to other common cancers. It would be helpful for them to use their data to try to shed insight into this.

Spatial analysis using Slide-seqV2 is important but mostly descriptive. It would be good to further highlight this work and how these findings could be actionable. It would be helpful to focus on cancer-immune crosstalk.

They developed a "new computational approach which regressed out context-dependent differences...". However, there was no validation of this algorithm to provide confidence that its output is accurate.

Line 211: they identified only one fibroblast subpopulation as compared to two endothelial and two pericyte subpopulations. Other studies have found multiple fibroblast subpopulations within other tumor types. Is homogeneous fibroblasts a feature unique to prostate?

Manuscript goes into great details describing the immunosuppressive TME of prostate cancers. This is largely known. It would be very helpful to push their data further to provide mechanistic insights

into how the immunosuppressive TME came about. Is it possible to identify tissue-resident macrophages?

Line 346: "expansion of exhausted CTLs" – can exhausted T cells expand?

Minor:

Use of 'normal' for adjacent tissues may not be accurate – more appropriate would be 'non-cancerous'.

Tumor-draining lymph nodes are often resected in prostate surgeries. It would be very interesting to do similar analyses on TDLNs.

Reviewer #3 (Remarks to the Author):

In the manuscript "Integrated single-cell and spatial transcriptomic analyses unravel the heterogeneity of the prostate tumor microenvironment", Hirz and colleagues applied single-cell genomics to uncover the tumor microenvironment (TME) properties of prostate adenocarcinoma.

The authors sampled patients with newly diagnosed prostate cancer with high grade and low grade cancer and, as a control, sampled tumor-adjacent gland tissues from most of these patients. As 'healthy' control the authors sampled 4 patients with bladder cancer undergoing removal of bladder and prostate and one patient with lung cancer. Furthermore, the authors applied spatial transcriptomics (SlideSeqV2) to spatially resolve the tissue architecture.

The authors segmented the spatial transcriptomics data into 'tumor-context' and 'tumor-adjacent-context' and devised a computational pipeline to correct for context-dependent admixture artifacts in the transcriptional comparison of cellular states between contexts. The authors used various gene signatures to dissect TME properties and utilized the spatial data for confirmation of findings. They discovered that stromal cells within tumors, and more specifically within the tumor context, seem to be highly angiogenic and suggested a role for them in remodeling the microenvironment of the tumor. They used their spatial information to infer ligand-receptor interactions between tumor and stromal cells that are supported by spatial vicinity. Within the immune cells, the authors found that MDSC-like monocytes, M2-macrophages, exhausted effector T-cells, and Treg cells are enriched or more active in tumor tissues, compared to healthy controls, normal tumor-adjacent tissues, and tumor-adjacent context. This highlights the immune-suppressive environment, importantly in low grade and high grade prostate adenocarcinoma.

In summary, the authors presented an unprecedented map of immune cells in the TME and the contextualized spatial information, including cell-cell interaction inference based on the spatial context. Furthermore, the correction of context-dependent admixture artifacts will be useful for others who seek to contextualize their spatial transcriptomics data. In summary, the presented manuscript will represent a valuable resource for the field with a large audience, although there are some points that should be revised.

Major comments:

1. In the Methods, all the technical aspects of 10x processing, slide-seq processing and sequencing are totally missing.
2. Looking at the map quality in Suppl Table 1, the mean read per cell varies from 2,500 to 200,000. How does this influence data processing and data integration with Conos? Do criteria have been applied to exclude cells or samples?
3. Line 101: it stands that 179,359 cells have been analyzed. When we sum Table S2-column 2 we find that 187,457 cells analyzed. Can the authors explain?
4. Why any neutrophil is found in the dataset?

5. Even though the SlideSeq data is used mostly as validation, much of the manuscript's novelty is based on these data. Nevertheless, only one patient (n=1) has been analyzed. Increasing the sample size, if in scope, could make observations more solid.

6. Reading 'Table S2-Cases_for_Slide_seq' is very confusing: what patient are we talking about? The median read per cell identified per Slide-seq is very low ranging from 400 to 650. The number of reads per cell is almost equal to the number of genes detected. Is that enough to call the cell types?

7. Suppl. Fig. 2C: Only a few samples seem to contribute substantially to the malignant cell population. Could this influence the DEGs identified as 'prostate tumor signature'? Could the signature be useful to discriminate between malignant and non-malignant cells in other malignancies with luminal-epithelial cell origin?

8. Line 169-174: the authors applied ICA analysis and obtained Figure S3A. It is really not clear how this analysis was applied and why only 6 samples have been selected. The cluster C1 shows a signature with FOS, JUNB. These genes come along with tissue dissociation artifacts as abundantly described before. Is it possible that tissue dissociation induces spurious gene signatures in addition to the tumor signature?

9. Figure 2H: Regarding the approach of linear regression to correct for context-dependent admixture : The authors do not show if their approach confers an advantage over only taking high confidence "pure" beads into account.

10. Cell population abundance comparison between sample types should take into account the proportional nature of population counts. Therefore, in addition to or instead of wilcoxon more specialized statistical models should be used, for example (10.1038/s41467-021-27150-6). Especially for M2 macrophages in this manuscript, this would be important. As is now, it could also be possible that antigen presenting macrophages are reduced in Adj-N and Tumor samples and the statistical test used finds M2-macrophages to be enriched, even though this could be an artifact of negative correlation.

11. The authors use various gene signatures to calculate signature scores. Where did the authors derive those signatures from?

12. In lines 349-351 the authors state that they assayed Treg activity as a surrogate for CD4+ T-cell function. As Tregs and other CD4+ T-cell subpopulations (e.g. CD4+ effector memory T-cells) have very different functions, Treg activity seems not like an appropriate surrogate for overall CD4+ T-cell function. Furthermore, Treg activity was not assayed. A score based on a gene signature was calculated, but no functional assay was conducted. Please also refer to 'Treg activity score' in Figures 6F and 6G to avoid confusion.

Minor comments:

1. Figure 1a can be enhanced to add a description of the cohort (how many patients, how many patients analyzed with scRNA-seq, how many with slide-seq, etc...).

2. Figure 1D: the color code should be defined in the legend

3. Figure 3D: Please add a legend indicating that red circles are tumor markers to make it easier visible what they mean (even though it is in the legend and supplementary info).

4. Typo: Figure 3G: Please correct 'Spatial Auto-correction' to 'Spatial Auto-Correlation'

5. Typo: Supplementary Figure 5B: it is 'pseudotime' not 'sudotime'

Overview

We thank the reviewers for carefully reading our manuscript and for their insightful comments. We have addressed in detail the specific comments by a substantial number of additional experiments, analyses and a revision of the main text and figures.

The gene lists that make up the gene signatures (scores) are listed in **Table S4**.

Key highlights in our revisions include:

- Utilizing public single cell RNA sequencing data from other cancer types to check the differences in T cell and myeloid cell gene expression patterns (functional characterization) comparing “cold” and “hot” tumors.
- Identifying CCL20-CCR6 axis as a mechanistic pathway involved in immune suppressive tumor microenvironment of prostate cancer.
- Comparing myeloid and T cell functional characterization between localized prostate tumor to prostate metastases including lymph nodes and bone.
- Validating the specificity of “Prostate Tumor Gene Signature”.
- Validation of tumor infiltration by M2-macrophages using compositional data analysis (CoDA) and cluster-free cell compositional analysis.

Reviewer #1

In this paper, the authors investigated the TME heterogeneity of localized prostate cancer with integrated single cell sequencing and spatial transcriptome technology. Their analysis revealed an immunosuppressive microenvironment associated with suppressive myeloid populations and T cell exhaustion, along with high stromal angiogenic activity. These findings could help us understand the heterogeneity of prostate cancer. However, we still have several concerns:

Major concern:

In general, this work seems descriptive but lacks novel findings. Prostate cancer is a well-known “cold” tumor, which contains few immune cells. Although the authors find many immunosuppressive cell types in the TME, most of these findings have already been reported in studies on other tumor models. These results still could not explain why prostate cancer is “cold”. And it also could not explain the difference between prostate cancer and other solid tumors.

We thank the reviewer for this observation. As pointed out, “cold” tumors manifest as a low density of tumor-infiltrating immune cells, in particular a low density of infiltrating T-cells.

For our single cell transcriptomic data, we used a method of tumor dissociation (enzyme cocktail, incubation time) that allowed us to enrich for, and to focus on immune cells (collagenases+Dispase). However, as this method is slightly different from other published studies, it is not possible to compare cell fractions from our dataset with other studies looking at different tumor types.

The Slide-Seq data that we provide preserves the entire tumor architecture of the prostate tissue (**Figure R1A**), and this did demonstrate the expected very few immune cells (**Figure R1B**) comparing Slide-seq data to scRNA-seq data using two different dissociation protocols (ours: Collagenases+Dispase and published protocol: Rocky).

Figure R1. A. (Figure S6C) High-resolution spatial representation using Slide-seqV2 to identify immune cells in prostate tissue. Here we show healthy prostate, adjacent-normal and two tumor tissues collected from low-grade (Tumor-LG) and high-grade (Tumor-HG) patients. Patients ID from Table S2 represented here as healthy is HP1, adj-normal of LG case is Benign04, tumor tissue of LG case is Tumor08, tumor tissue of HG case is Tumor02. **B.** Stacked barplots showing the fractional composition of cell number for different clusters within Slide-seqV2.

As it was not possible to compare cell fractions, instead we compared the functional status of different immune cell subsets from prostate cancer (PCA) to those seen in other solid tumors. We compared “cold” tumors including PCA and pancreatic ductal adenocarcinoma (PDAC) (1) to “hot” tumors such as Head and Neck squamous cell carcinoma (HNSCC) (2), liver hepatocellular carcinoma (LIHC) (3) and lung cancer (lung) (4).

Regarding the T cell transcriptional state, we performed an integration for T cell compartments from all datasets mentioned above where data showed the three CD8+ T cell subsets (CTL-1, CTL-2 and CD8+ effector cells) obtained in our dataset also in the other solid tumors (**Figure R2A**). We checked the cytotoxicity gene expression pattern in these CD8+ T cell subsets among the five different cancer types. Data in **Figure R2B** show that “cold” tumors, including PCA and PDAC, have significantly lower cytotoxicity scores in their CD8+ T cell subsets compared to the “hot” tumors (lung, LIHC and HNSC).

Figure R2. A. Joint embedding showing the detailed annotation of lymphoid subpopulations in different cancer types including prostate cancer (PCA), pancreatic ductal adenocarcinoma (PDAC), Head and Neck squamous cell carcinoma (HNSCC), liver hepatocellular carcinoma (LIHC), lung cancer (lung) and join (all samples). **B.** Boxplots represent the average expression of cytotoxicity score in CD8+ T cell subsets (CD8+ effector cells, CTL-1 and CTL-2) in different cancer types (PCA, PDAC, Lung, LIHC, HNSC). Statistics are accessed with Wilcoxon rank sum test (* $p < 0.05$, ** $p < 0.01$, *** $p < 0.001$, **** $p < 0.0001$).

Checking the exhaustion score, we compared CD8+ effector cells and CTL-1 with high exhaustion score from our dataset (**Figure R3A**) to other tumor types. We did not see any main differences comparing “cold” to “hot” tumors (**Figure R3B**). Similarly, no main differences between “cold” and “hot” tumors were obtained at the level of Treg activity (**Figure R3C**).

Figure R3. A. (Figure 6C) Boxplots represent the average expression of exhaustion score in CD8+ T cell subsets (CD8+ effector cells, CTL-1 and CTL-2) in our dataset of prostate cancer. **B-C.** Boxplots represent the average expression of exhaustion score in CD8+ T cell subsets (CD8+ effector cells, CTL-1 and CTL-2) (**B**) and Treg activity (**C**) in the 5 different cancer types (PCA, PDAC, Lung, LIHC, HNSC). Statistics are accessed with Wilcoxon rank sum test (* $p < 0.05$, ** $p < 0.01$, *** $p < 0.001$).

At the same time, we examined the immunosuppressive myeloid population. We performed an integration for myeloid compartments from our dataset (PCA) compared to other solid tumors where we observed the M2-macrophages (Macrophages 3) as well as tumor inflammatory monocytes (Mono2) (**Figure R4A**) which showed high expression for myeloid derived suppressor cells (MDSCs) in our dataset (**Figure R4B**).

Figure R4. A. Joint embedding showing the detailed annotation of myeloid subpopulations in different cancer types including prostate cancer (PCA), pancreatic ductal adenocarcinoma (PDAC), Head and Neck squamous cell carcinoma (HNSCC), liver hepatocellular carcinoma (LIHC), lung cancer (lung) and join (all samples).

B. (Figure 5D) Boxplot comparing the average expression of MDSC gene signature in Mono2 annotated as tumor inflammatory monocytes (TIMo) across the three different prostate samples (healthy, adjacent normal and tumor). Boxplots include centerline, median; box limits, upper and lower quartiles; and whiskers are highest and lowest values no greater than 1.5x interquartile range. Statistical significance was accessed using Wilcoxon rank sum test (* $p < 0.05$, ** $p < 0.01$).

We compared the M2 gene signature pattern in Macrophages-3 (**Figure R5A**) and MDSC score in tumor inflammatory monocytes (Mono2) (**Figure R5B**) among the five different cancer types. Data showed no significant differences comparing “cold” to “hot” tumors; however, data showed more

enrichment of M2 like macrophages (Macrophages 3) in prostate cancer compared to the other cancer types (**Figure R5C**).

Figure R5. A-B. Boxplots represent the average expression of M2 score in Macrophages 3 (**A**) and MDSC score in tumor inflammatory monocytes (Mono2) (**B**) in different cancer types (PCA, PDAC, Lung, LIHC, HNSC). **C.** Boxplot representing the cell fraction of different myeloid subpopulations across the 5 different cancer types (PCA, PDAC, Lung, LIHC, HNSC). Statistical significance was accessed using Wilcoxon rank sum test (* $p < 0.05$, ** $p < 0.01$).

Based on the following data, it seems that we do have less cytotoxic functional T cells in the “cold” tumors compared to “hot” tumors with no significant differences at the level of myeloid immunosuppressive gene expression patterns.

This point has been added to the main manuscript in the “**Results**” and “**Discussion**”.

Results:

Line 484:

“Also, we aimed to check the CTLs cytotoxicity by comparing “cold” tumors characterized with low T-cell infiltration and “hot” tumors with high degree of T-cell infiltration (84). To this end, we collected scRNA-seq data from pancreatic ductal adenocarcinoma (PDAC) (87) as another cold tumor; and from “hot” tumors including head and neck squamous cell carcinoma (HNSCC) (88), liver hepatocellular carcinoma (LIHC) (86) and lung cancer (lung) (4). We first performed an integration for T-cell compartments from all datasets mentioned above where data showed the three CD8+ T-cell subsets (CTL-1, CTL-2 and CD8+ effector cells) obtained in our dataset also in the other solid tumors (**Figure S6B**). Interestingly, T-cell cytotoxicity scored significantly higher in the three different CTLs in the “hot” tumors compared to “cold” tumors (**Figure 6E**) that affects the efficacy of immune therapy.”

Discussion:

Line 857:

“Comparing the T-cell cytotoxicity of “cold” and “hot” tumors showed significantly higher cytotoxicity score in the latter (Figure 6E), suggesting more functional T-cells that may have an effect on tumor response to immune therapy. No differences were obtained comparing the immune suppressive myeloid compartment between “cold” and “hot” tumors such as MDSC score in TIMO and M2 score in M2-macrophages (data not shown).”

Minor concern:

1. Figure 1D, color of this bar plot is really confusing. Annotation should be added.

The colors have been corrected and a figure legend has been added (**Figure 1D**).

Figure R6. (Figure 1D) Stacked barplots showing the fractional composition of cell number for different clusters within scRNA-seq (using two different dissociation protocols: Collagenases+Dispace and Rocky) and Slide-seqV2.

2. Stroma cells annotated as pericyte by the authors contain two different subpopulations. It may be able to be divided into pericyte and SMC with conventional markers, such as MYH11. And as shown in Figure 1B, it seems like that one population was enriched in tumor tissues. However, it's difficult to know which one was that population in the current result.

Thank you. The pericyte-2 subpopulation expressed a gene signature enriched for a transcriptional program of muscle contraction consistent with vascular smooth muscle.

Line 337: “The expression pattern in Pericyte-1 cells was enriched for pathways involved in extracellular structure organization and connective tissue development, while Pericyte-2 cells demonstrated gene signatures enriched for muscle contraction consistent with vascular smooth muscle cells (VSMCs) (Figure 3B).”

To expand upon this finding, we checked for the expression of several smooth muscle cell markers such as MYH11 (as suggested) in addition to alpha smooth muscle actin (Acta2) (5), Desmin (Des) (6), MCAM (7), and Calponin CNN1 (8,9). As shown in **Figure R7 (Figure S4A)**, pericyte-2 subpopulation showed high expression for these genes compared to pericytes-1.

Figure R7. (Figure S4A) Dotplot representing key-marker gene expression in stroma subpopulations. The color represents scaled average expression of marker genes in each cell type, and the size indicates the proportion of cells expressing marker genes.

Based on the reviewer’s suggestion, we added the pericytes annotation to **Figure R8A (Figure 1B)**. Then we checked the cell abundance for each pericyte subcluster comparing the three different fractions (healthy, Adj-normal and tumor). Data showed a slight but non-significant increase in pericytes-1 and pericytes-2 abundance in tumor fractions compared to both healthy and adj-normal (**Figure R8B**).

Figure R8. A. (Figure 1B) UMAP visualization of major cell populations highlighting annotation of different stromal subpopulations in the joint embedding. **B.** Comparison of pericytes-1 and pericytes-2 abundance (fraction of total cells per sample) shown as boxplot. Statistical significance was accessed using Wilcoxon rank sum test.

3. Line 283, the expression should be modified. “Immunosuppressive myeloid cells are enriched in prostate tumors” may be better.

Thank you. The sentence has been changed.

Line 401: “Immunosuppressive myeloid cells are enriched in prostate tumors”

4. The hyperlink in Data Availability only contains 3’ single cell sequencing data, while slide-seq V2 data was not uploaded.

We have uploaded Slide-seq V2 data and corresponding cell annotations to GSE181294 (S1-S12).

5. Page 50. It’s difficult to understand the meaning of the figure legend of Figure S7. Was there a writing mistake?

We have corrected this mistake in the figure legend of Figure S7.

“The prostate cancer TME is enriched by exhausted CD56^{DIM} NK cells and activated B cells.”

Reviewer #2**Major comments:**

Their introduction starts with the important contrast between indolent versus aggressive prostate cancer. In their study, they segregated tumors into high grade (HG) versus low grade (LG). While HG vs. LG and aggressive vs. indolent are highly related, they are not the same thing [PMID: 24027026]. It would be very useful if their data can shed new insights into differences between indolent versus aggressive disease. Rather, many of their findings could not differentiate high-grade versus low grade tumors, let alone indolent versus aggressive.

We thank the reviewer for this important remark. Since low Gleason score prostate tumors can be distinguished as indolent and aggressive subgroups (based on gene expression associated with aging and senescence) and high-grade vs. low-grade are not the same thing as indolent vs. aggressive, we removed the words “indolent” and “aggressive” as identification for low-grade and high-grade, respectively, from our introduction.

Line 73: “Localized prostate cancer is a clinically heterogeneous disease. Some patients present with low-risk prostate tumors that can safely be observed, while others have high-risk disease that carries a substantial relapse risk even following state-of-the-art treatment.”

We did not identify significant differences when comparing low-grade and high-grade cases (for example, immunosuppressive gene expression patterns for myeloid and T cell subsets) (**Figure R9**). We suspect that this is likely due to the low number of samples in each fraction (healthy samples n=5, adjacent-normal samples n=9 from low-grade cases and n= 5 from high-grade cases, tumor samples n=8 for low-grade cases and n=5 for high-grade tumors).

As suggested by the reviewer, we checked as well for the expression of the 19-gene signature (10) in our dataset. Notably, some of these genes to be highly expressed in immune cells (ITM2A, NFE2L2, CTSH, MSN) and other genes to be highly expressed in non-immune stromal subsets (GPX3, CDKN1A, PMP22, SERPING1, LGALS2, FGFR1) (**Figure R10A**). Based on this, we examined whether the expression of these genes can differentiate our tumor samples, especially the LG ones, as “indolent” and “aggressive” samples; however, we did not such differentiation among our tumor samples (**Figure 10B and C**).

Figure R9. Boxplots represent the average expression of exhaustion score in CD8+ effector cells and CTL-1 subsets, and MDSC score in tumor inflammatory monocytes (Mono2) in five different fractions including healthy prostate tissues (Healthy), adjacent normal prostate tissues collected from low-grade cases (Adj-N LG) and high-grade cases (Adj-N HG) and tumor prostate tissues collected from low-grade cases (Tumor-LG) and high-grade cases (Tumor-HG). Boxplots include centerline, median; box limits, upper and lower quartiles; and whiskers are highest and lowest values no greater than 1.5x interquartile range. Statistical significance was accessed using Wilcoxon rank sum test (*p<0.05, ***p<0.001).

B.

C.

Figure R10. A. Dotplot representing the expression of 19-genes from Irshad et al. in our dataset. The color represents scaled average expression of marker genes in each cell type, and the size indicates the proportion of cells expressing marker genes. **B-C.** Expression of selected genes from the 19-gene signature in immune cell subsets (**B**) and non-immune stromal subsets (**C**) visualized in violin plot comparing tumor samples collected from low-grade (LG) cases and high grade (HG) cases.

Another important point in their introduction is “The prostate TME typically contains few immune cells...” as compared to other common cancers. It would be helpful for them to use their data to try to shed insight into this.

Thank you. Prostate tumors have been described as poorly immunogenic. With this in mind, we devised a dissociation protocol to allow for enrichment and preservation of the immune populations. Our data, as well as that of others (11,12) showed that different dissociation protocols affect cell yield in solid tumor tissues as shown in **Figure R11** comparing cell fraction of immune cells versus non-immune cells (fibroblasts, pericytes, endothelial cells, epithelial cells) using our dissociation protocol (collagenase+dispase) versus Rocky (11).

As expected, our spatial transcriptomic analysis (Slide-seqV2, preserved architecture, no tissue dissociation) showed few immune cells compared to non-immune stroma and epithelial cells (**Figure R11**). This means that we need to use data from dissociation-free spatial analysis to compare cell fraction of immune compartments in our dataset of prostate cancer to other cancer types.

Figure R11. (Figure 1D) Stacked barplots showing the fractional composition of cell number for different cell populations within scRNA-seq (using two different dissociation protocols: Collagenases+Dispase and Rocky) and Slide-seqV2.

Due to the low number of Slide-seqV2 samples, we used the data we obtained from multiplex immunohistochemistry (mIHC). mIHC performed *in-situ* on the same tissue samples as the single-cell expression. We used an immune panel including six antibodies panel consisting of CD3, CD8, PD-1, FOXP3, CD68, CD163, along with DAPI counterstaining.

Our data showed low percentage of T cells in stromal (12.71%) and intra-epithelial (1.82%) regions in prostate cancer (**Figure R12A**) compared to metastatic melanoma (**Figure R12B**) (13) and breast cancer (14) (**Figure R12C**). Data in both metastatic melanoma and breast cancer showed higher percentages in CD8+ T cells and CD4+ T cells in both tumor regions (stroma and intra-tumoral) compared to percentage of total T cells obtained in our dataset (primary prostate cancer).

Figure R12. **A.** Data for all primary prostate tumor samples in our study cohort including the % of T cells (CD3+) in two tumor regions (intra-tumoral (IT) and stromal regions (S)) using mIHC. **B.** Data showing percentage of T cell subsets (CD8+ and CD4+) in three tumor regions (intra-tumoral (IT), tumor margin (TM) and stromal regions (S)) as well as the average of these regions collected from human metastatic melanoma using mIHC. Statistical analysis (Mann Whitney) was performed across the cohort, significantly different data is represented by *($p < 0.05$) and ***($p < 0.005$) (Halse et al., Sci rep 2018). **C.** Data showing the percentage of different immune cells collected from human breast cancer samples in stroma, tumoral and tertiary lymphoid structure (TLS) using mIHC.

Spatial analysis using Slide-seqV2 is important but mostly descriptive. It would be good to further highlight this work and how these findings could be actionable. It would be helpful to focus on cancer-immune crosstalk.

We thank the reviewer for pointing out this shortcoming. Slide-seqV2 and in general, spatial transcriptomics, holds the potential to reveal characteristics that are otherwise not observable using only single-cell transcriptomics assays. Firstly, we provided a single cell spatial map of prostate cancer tumor microenvironment. We have shown that the spatial organization of the tumor cells can be identified by the tumor-specific signature identified in our single-cell RNA-seq dataset. This is an important validation since we have identified a gradual loss in structure (captured by Moran's I score in **Figure R13**) in the growing tumor. This characterization has helped us to delineate tumor-infiltrated region versus the non-infiltrated region. Tumor-infiltrated region has been segmented to demarcate the local context within the puck where the tumor infiltration has taken place. Cell-cell interaction plays a crucial role in shaping the structure of the tumor microenvironment. However, while extracting the significant ligand-receptor pairs (LR), one must be cognizant of the spatial occurrence of the actual cells. Therefore, we used the spatial information to assign significance

scores to the LR pairs. From a list of almost ~1200 LR interactions, we have identified 405 significant communication channels. Here we have concentrated on tumor-stromal interaction due to the very few immune cells in the Slide-seqV2 dataset. However, we agree that in the tumors enriched with immune cells, the cancer-immune crosstalk will be one of the most important attributes to study. We note that the computational methods that are showcased in our manuscript can be directly applicable in such cases.

Figure R13. (Figure 1H) Barplots showing spatial autocorrelation (Moran's I) of fibroblasts and pericytes in Healthy (n=4), adj-Nomral (n=4) and Tumor samples (n=4). Moran's I evaluates whether the cells are clustered (high Moran's I score) or dispersed (low Moran's I score). Statistical analysis was performed using Wilcoxon rank sum test. (*p<0.05, error bars: SEM).

They developed a “new computational approach which regressed out context-dependent differences...”. However, there was no validation of this algorithm to provide confidence that its output is accurate.

Thank you. It is indeed difficult to measure the accuracy of such regression-based approach. To the best of our knowledge, currently there are no spatial simulator that can provide the ground truth to test the proposed algorithm. However, we resorted to the functional validation to test the prediction. First, we obtained an overexpression of tumor markers in non-tumor cellular population, which is likely a cause of spatial diffusion. The corrected gene expressions from our algorithm were able to suppress this overexpression. On the other hand, we compared the results of gene ontology (GO) data analysis we obtained for Endothelial-2 from Slide-seqV2 data and scRNA-seq data (10X). Data showed similar results in the enriched pathways in Endothelial-2 comparing Slide-seqV2 (**Figure 14A**) and 10X scRNA-seq data (**Figure 14B**) such as cell migration and angiogenesis. This was also validated using Moran's I data analysis which evaluates the extent to which the cells are clustered (high score) or dispersed (low score) (**Figure 14C**).

Figure R14. **A.** Dotplot shows GO terms of upregulated genes in Endothelial-2 cells in tumor context compared to tumor-adjacent context. **B.** Enriched GO Biological process (BP) terms of top 300 upregulated genes for Endothelial-2 comparing tumor to adjacent-normal. Statistical analysis was performed by over-representation test. **C.** Comparison of spatial autocorrelation (Moran's I) of Endothelial-2 in healthy (n=4), adj-normal (n=4) and tumor samples (n=4).

Together, this functional validation provides us the confidence that our algorithm can be useful for regressing out the effect of linear diffusion. We have also added a toy simulation to judge the effectiveness of the regression-based correction mechanism.

In the attached [notebook \(https://github.com/hiraksarkar/Prostate_Context_Paper/blob/main/Simulating_DE_prostate_paper.ipynb\)](https://github.com/hiraksarkar/Prostate_Context_Paper/blob/main/Simulating_DE_prostate_paper.ipynb), we show how the regression based correction suppresses context-specific markers.

Here we describe the main steps for the simulation:

1. We collected bulk RNA-seq samples that correlate (Pearson correlation of ~ 0.85) well with each other, that are representative of two different cell types. For ease of understanding, we named them as stromal cell-type and tumor cell-type.
2. We use these bulk transcriptomic expression profiles as the source of the base context and produced a linear mixture of to emulate the spatial mixture, but the proportion of mixture is skewed towards the stromal profile. We called this mixture profile “non-tumor context”.
3. Next, we perturbed the base bulk RNA-seq profiles to introduce differential expression (DE) genes and then simulated an ad-mixture profile where proportion of tumor cell-types are high, producing “tumor context”.
4. We first identified if tumor DE genes are overexpressed in stromal cells where we quantified DE genes using EdgeR. Consequently, we used regression-based correction mechanism. Below are the plot showing the tumor DE genes getting corrected (in red circles), highlighting the stromal DE genes (**Figure 15**).

Figure 15. From the toy simulation, the Z values for the fold changes obtained from running EdgeR DE tool on stromal profiles between not-tumor and tumor context are plotted. The regression-based correction was able correct (y-axis) for the tumor-DE genes that are otherwise marked significant (in x axis) by the DE method EdgeR when run on uncorrected profiles. On the other hand the actual DE genes are still significantly high (marked in green).

Line 211: they identified only one fibroblast subpopulation as compared to two endothelial and two pericyte subpopulations. Other studies have found multiple fibroblast subpopulations within other tumor types. Is homogeneous fibroblasts a feature unique to prostate?

We thank the reviewer for this important remark regarding the fibroblasts. Cancer-associated fibroblasts (CAFs) are felt to be a key component of the tumor microenvironment. CAFs may have additional subpopulations and diverse functions as has been described in various tumor types.

To check whether homogeneous fibroblasts is a feature unique to prostate, we extracted the fibroblasts from our dataset and re-ran data integration and clustering. A single cluster was identified, we did not observe multiple fibroblasts subpopulations (**Figure R16**).

Figure R16. UMAP embedding of fibroblasts from our dataset of prostate cancer showing expression of fibroblasts key marker genes (DCN and LUM).

We checked the expression pattern of key marker genes for different types of fibroblasts obtained in other tumors including inflammatory CAFs (iCAFs) and myo-CAFs (mCAFs) in bladder cancer (15) as well as vascular CAFs (vCAFs) in breast cancer (16), checking whether we can identify “multiple” fibroblasts based on their expression for these genes. Fibroblasts from our dataset express genes of both iCAF and mCAF as homogenous population but not vCAF (**Figure R16**), suggesting one fibroblast population in our dataset.

We then prepared a heatmap looking at signature genes described in the mCAF, iCAF, and vCAF subpopulations. Here there were subtle differences that seemed to suggest the clear presence of mCAF and iCAF populations, though no obvious separate vCAF signature.

This data is presented below in two formats (gene expression in UMAP (**A**) and heatmap (**B**) as shown below in **Figure R17**.

A.

mCAF gene markers:

iCAF gene markers:

vCAF gene markers:

B.

Figure R17. A. UMAP embedding of fibroblasts showing expression of mCAF, iCAF and vCAF key marker genes. **B.** Heatmap showing average gene expression of mCAF, iCAF and vCAF signatures in fibroblasts.

Manuscript goes into great details describing the immunosuppressive TME of prostate cancers. This is largely known. It would be very helpful to push their data further to provide mechanistic insights into how the immunosuppressive TME came about. Is it possible to identify tissue-resident macrophages?

Thank you. To provide mechanistic insights that may lead to immunosuppressive TME, we identified ligands that are highly expressed in myeloid cells as well as receptors that are upregulated in the lymphoid compartment of the tumor fraction. Then we examined the potential ligand-receptor interaction between the myeloid and T cell compartments.

This resulted in the identification of different significant potential channels of interaction including signaling through the CCL20-CCR6 axis (**Figure R18A**). CCL20 was most highly expressed by Mono2 (tumor Inflammatory monocytes (TIMo)) which showed significantly higher expression of Myeloid Derived Suppressor Cells score (MDSC score) in tumor fraction (**Figure R18B**), and the CCR6 was upregulated in different T cell populations including Tregs which showed significantly higher Treg activity in tumor fraction compared to adjacent normal and healthy prostate samples (**Figure R18C**).

To examine whether CCL20-CCR6 axis is involved in immune suppressive TME of prostate cancer, we injected parental RM1 prostate cancer cell line subcutaneously (0.25×10^6 cells) into syngeneic mouse (C57Bl6 wild-type male mice) and then divided the mice into three different groups (5 mice/group):

1. Group1: treated with isotype antibodies (IgG1 + IgG2a).
2. Group 2: treated with anti-CCL20 to block the interaction between CCL20 (myeloid cells) and CCR6 (T cells) + IgG2a (isotype antibody control for anti-PD1).
3. Group 3: combination of anti-CCL20 + anti-PD1.

The treatment started on day 7 with anti-CCL20 200 ug/kg and anti-PD1 (6 mg/kg) every 3 days for a total of 3 times. No expression for CCL20 and CCR6 was obtained on RM1 prostate cancer cell line (**Figure 18D**).

Data showed the anti-CCL20 treatment to significantly reduce RM1 tumor growth, though there was no combinatorial effect with the addition of anti-PD1 therapy (**Figure R18E**).

Figure R18. A. (Figure 7B) A computational approach using single-cell transcriptomic data highlighting CCL20-CCR6 interaction between myeloid and T cell subsets. **B. (Figure 7C)** Boxplot comparing CCL20 expression in different myeloid subsets (left) and comparing the average expression of MDSC gene signature in Mono2 annotated as tumor inflammatory monocytes (TIMo) across the three different samples (Healthy, adjacent normal and tumor) (right). Statistical significance was accessed using Wilcoxon rank sum test (* $p < 0.05$, ** $p < 0.01$, *** $p < 0.001$, **** $p < 0.0001$). **C. (Figure 7D)** Boxplot comparing CCR6 expression in different T cell subsets (left) and comparing the average expression of Treg activity gene signature in Treg subpopulation across the three different samples (Healthy, adjacent normal and tumor) (right). **D.** For CCL20 and CCR6 expression in RM1 cells, 500-1000 cells were sorted into a 96 well plate and Smartseq2 protocol was used to measure gene expression. RM1 cells were FACS sorted from subcutaneous tumors in mice based on reporter expression ($n=4$, 3 technical replicates per mouse). **E. (Figure 7E)** Mice (5 mice/group) were injected subcutaneously with 0.25×10^6 RM1 cells. anti-CCL20 (1.8mg/kg) and/or anti-PD1 (6mg/kg) were injected intraperitoneally to mice every 3 days for a total of 4 times. Tumor growth was monitored by caliper measurement of the tumor volume (* $p < 0.05$).

Text describing the above results has been added to the main manuscript in addition to adding the panels of **Figure R14** to the main figures (**Figure 7**).

Results:

Line 655: Based on this correlation between myeloid and T cell suppressive populations, we checked the ligands that are highly expressed in TIMo and receptors that are upregulated in the Tregs in the tumor fraction, followed by significant potential ligand-receptor interactions between the two subpopulations. Data showed CCL20-CCR6 as one of the significant axes of interaction between TIMo and Tregs, respectively (**Figure 7B**). CCL20 was highly expressed in TIMo and its cognate receptor CCR6 was predominantly expressed in Tregs, Th1 and Th17 (**Figure 7C**, **Figure 7D**). Several studies showed an effect for CCL20 signaling in enhancing tumor growth, invasiveness, and chemoresistance (52,94,95) by recruiting Tregs and/or Th17 (96,97).

To examine whether CCL20-CCR6 axis is involved in the immune suppressive TME of prostate cancer, we injected parental RM1 prostate cancer cell line subcutaneously into C57BL/6J wild-type (WT) mice. When tumor reached the volume of $300-400 \text{ mm}^3$, mice were treated with CCL20-

blocking antibody alone or in combination with immune check point blockade (anti-PD1). Blocking CCL20-CCR6 interaction using CCL20-blocking antibody reduced significantly RM1 tumor growth with no combinatorial effect for ant-PD1 (**Figure 7E**), suggesting CCL20-CCR6 axis to be involved in the prostate immune suppressive TME.”

Discussion:

Line 854: “Through computational heuristics, we identified CCL20-CCR6 as a potential ligand-receptor axis that may allow for communication between TIMo and Tregs. We showed that blocking one such signaling axis using CCL20 blocking antibody significantly reduced tumor growth in a subcutaneous model of syngeneic prostate cancer. This axis is implicated in several inflammatory and immune-activated states, including autoimmune disease (113). The potential for modulating the axis to reduce the activated states of immune cells has been extensively explored and led to early-stage clinical trials (114,115).”

Methods:

Line 1128: “CCL20 blocking antibody treatment

RM1 prostate cancer cells (0.25×10^6 cells) were injected subcutaneously to C57BL6/J male mice (#000664) from The Jackson Laboratory. When tumor reached the volume of 300-400 mm³, mice received an intraperitoneal injection of 45ug anti-CCL20 blocking antibody (R&D Systems, clone 114908) or rat IgG isotype control antibody (R&D Systems, clone 43414), 150ug of anti-mouse PD1 (BioXcell, clone RMP1-14) or rat IgG2a isotype control (BioXcell, clone 2A3). The mice were treated every 3 days for a total of 3 times. Tumor growth was monitored by caliper measurement of the tumor volume.

RM1 cells were maintained in DMEM (Corning, 15-013-CV) complemented with 10% FBS (GIBCO by Life Technologies, A31605-01) and 1% Penicillin-Streptomycin (GIBCO by Life Technologies, 15140-122). All mice were maintained in pathogen-free conditions and all procedures were approved by the institutional Animal Care and Use Committee of Massachusetts General Hospital.”

In addition, we checked for the expression of tissue-resident key marker genes such as SEPP1 and FOLR2 which showed high expression in one of the macrophages subsets (**Figure R19**) (17).

Figure R19. UMAP embedding of myeloid cells showing expression of tissue-resident macrophages key marker genes.

Line 346: “expansion of exhausted CTLs” – can exhausted T cells expand?

We appreciate this nuanced question. In fact, some studies showed a selective expansion for a subset of exhausted CD8 T cells “less exhausted T cells” by anti PD-L1 (18,19). However, in our study we did not have anti-PD1 pre-treated samples as well as no identification for this specific subset. We aimed to mention that we obtained more cytotoxic T cells (CTLs) in the tumor fraction that became dysfunctional, and eventually acquired an “exhausted” phenotype, compared to healthy prostate tissue.

We have corrected the text:

Line 514: “Measurement of T-cell abundance showed a higher proportion of exhausted CTL-1 cells in tissues collected from cancerous prostate compared to healthy prostate tissues (Figure S6F), suggesting that more CTLs in the tumor fraction became dysfunctional, and eventually acquiring an “exhausted” phenotype.”

Minor:

Use of ‘normal’ for adjacent tissues may not be accurate – more appropriate would be ‘non-cancerous’.

We appreciate the reviewer's suggestion, and we had similar thoughts as we were putting together the figures. However, we opted to use adjacent normal (Adj-N) in the text to align with the figures, mostly for the practical reason that “Adj-N” occupies less space compared to “Adj-non-cancerous”.

Also, we clarified in the text that adjacent normal “Adj-N” are the adjacent non-tumor prostate tissues.

Line 127: “Live, non-erythroid cells (DAPI^{neg}/CD235^{neg}) were collected by fluorescence-activated cell sorting (FACS) from healthy prostate tissues (n=5), prostate tumor tissues (n=12 LG and n=7 HG) and adjacent non-tumor prostate tissues (n=11 LG and n=4 HG, hereafter ‘adjacent-normal’ (Adj-N)).”

Tumor-draining lymph nodes are often resected in prostate surgeries. It would be very interesting to do similar analyses on TDLNs.

We agree that this is an interesting comparison. We queried public scRNA-seq data of lymph nodes with prostate cancer metastasis where we found one paper (15) where they worked on three prostate metastatic lymph nodes (LN) (GSE141445). In addition, we queried our data collected from prostate spine metastasis (BMET) (20) as another type of prostate metastasis to see the differences at the level of the immune microenvironment.

Since the method and the conditions of tumor dissociation (enzyme cocktail used, incubation time) affects cell yield, comparing the cell fraction from these three different datasets was not possible as the samples were not dissociated using the same protocol. We therefore compared the functional status of different cell subsets with a focus on checking for the immunosuppressive cells whether there are differences between the primary tumor site and its metastatic sites (LN and BMET).

We examined the myeloid population in prostate metastatic lymph nodes and spine where macrophages subsets showed high expression for M2 key marker genes such as CD163 and TREM2 (Figure R20A and B).

Prostate metastatic lymph nodes (LN)

A.

Prostate spine metastasis (BMET)

B.

Figure R20. A-B Joint embedding showing the detailed annotation of myeloid subpopulations in prostate metastatic lymph nodes (LN) (A) and prostate spine metastasis (BMET) (B).

Then we checked the differences in M2 gene expression pattern in tumor associated macrophages among the three different tumor fractions. Interestingly, data showed higher M2 score in both metastatic sites compared to the primary tumors (Figure R21A). Also, the data showed a higher MDSC score in Mono2 annotated as tumor inflammatory monocytes (TIMo) in metastatic sites compared to the primary tumors (Figure R21B). This data suggests that metastatic sites such as spine and lymph nodes have more immunosuppressive tumor microenvironment compared to the primary site.

Figure R21. A-B. Boxplots represent the average expression of M2 score in Macrophages 3 (A) and MDSC score in tumor inflammatory monocytes (Mono2) (B) in different cancer types (BMET, LN, PCA).

We also checked for T cell exhaustion gene signature in CTL-1 and CD8+ effector cells comparing the three different tumor fractions and interestingly, data showed higher exhaustion gene expression pattern in both CTLs subsets in prostate metastatic lymph nodes compared to primary prostate cancer but not bone metastatic cancer (**Figure R22**).

Figure R22. A. Joint embedding showing the detailed annotation of lymphoid compartments in prostate metastatic lymph nodes (LN), primary prostate cancer (PCA) and prostate spine metastasis (BMET). **B.** Boxplots represent the average expression of exhaustion score in CD8+ effector cells (**left**) and in CTL-1 subset (**right**) in different cancer types (BMET, LN, PCA). Statistics are assessed with Wilcoxon rank sum test (* $p < 0.05$, ** $p < 0.01$, *** $p < 0.001$).

In conclusion, it appears that the lymph node metastatic prostate tumor microenvironment has more immunosuppressive characteristics compared to primary prostate cancer. This interesting point needs to be further explored.

We added this point to the “**Discussion**”:

Line 869: “We also compared the immune suppressive TME obtained in our dataset of primary prostate cancer, with prostate metastatic bone cancer from our group (113), and prostate metastatic lymph node cancer (117). Data (not shown) suggested that the immune TME in metastatic sites is further immunosuppressed compared to primary prostate cancer, especially at the lymph node site, which may help guide immunotherapeutic strategies for metastatic prostate cancer.”

Reviewer #3

In the manuscript "Integrated single-cell and spatial transcriptomic analyses unravel the heterogeneity of the prostate tumor microenvironment", Hirz and colleagues applied single-cell genomics to uncover the tumor microenvironment (TME) properties of prostate adenocarcinoma.

The authors sampled patients with newly diagnosed prostate cancer with high grade and low grade cancer and, as a control, sampled tumor-adjacent gland tissues from most of these patients. As 'healthy' control the authors sampled 4 patients with bladder cancer undergoing removal of bladder and prostate and one patient with lung cancer. Furthermore, the authors applied spatial transcriptomics (SlideSeqV2) to spatially resolve the tissue architecture.

The authors segmented the spatial transcriptomics data into 'tumor-context' and 'tumor-adjacent-context' and devised a computational pipeline to correct for context-dependent admixture artifacts in the transcriptional comparison of cellular states between contexts. The authors used various gene signatures to dissect TME properties and utilized the spatial data for confirmation of findings. They discovered that stromal cells within tumors, and more specifically within the tumor context, seem to be highly angiogenic and suggested a role for them in remodeling the microenvironment of the tumor. They used their spatial information to infer ligand-receptor interactions between tumor and stromal cells that are supported by spatial vicinity. Within the immune cells, the authors found that MDSC-like monocytes, M2-macrophages, exhausted effector T-cells, and Treg cells are enriched or more active in tumor tissues, compared to healthy controls, normal tumor-adjacent tissues, and tumor-adjacent context. This highlights the immune-suppressive environment, importantly in low grade and high grade prostate adenocarcinoma.

In summary, the authors presented an unprecedented map of immune cells in the TME and the contextualized spatial information, including cell-cell interaction inference based on the spatial context. Furthermore, the correction of context-dependent admixture artifacts will be useful for others who seek to contextualize their spatial transcriptomics data. In summary, the presented manuscript will represent a valuable resource for the field with a large audience, although there are some points that should be revised.

Major comments:

1. In the Methods, all the technical aspects of 10x processing, slide-seq processing and sequencing are totally missing.

We have revised the Methods section and added the following information:

Line 969:

“Massively parallel scRNA-seq processing

Single cells were encapsulated into emulsion droplets using Chromium Controller (10x Genomics). scRNA-seq libraries were constructed using Chromium Single Cell 3' v2 Reagent Kit according to the manufacturer's protocol.

Briefly, the volume of the collected samples after sorting was decreased and the cells were examined and counted under a microscope with a hemocytometer. Cells then were loaded in each channel with a target output of average 4,000 cells. Reverse transcription and library preparation was done on C1000 Touch Thermal cycler with 96-Deep Well Reaction Module (Bio-Rad). Amplified cDNA and final libraries were evaluated on an Agilent BioAnalyzer using a High Sensitivity DNA Kit (Agilent Technologies). Individual libraries were diluted to 4nM and pooled for sequencing. Pools were sequenced with 75 cycle run kits (26bp Read1, 8bp Index1 and 55bp Read2) on the NextSeq 500 Sequencing System (Illumina) to 70-80% saturation level.

Line 1148:

“Slide-seqV2 processing and sequencing

Slide-seq arrays were prepared and spatial bead barcodes sequenced following Slide-seqV2 (13) protocol using arrays created with custom synthesized barcoded beads

(5’-

TTT_PC_GCCGGTAATACGACTCACTATAGGGCTACACGACGCTCTTCCGATCTJJJJJJJ
TCTTCAGCGTTCCCGAGAJJJJJNNNNNNNVVT30-3’)

with a photocleavable linker (PC), a bead barcode sequence (J, 14 bp), a UMI sequence (NNNNNNNVV, 9 bp), and a poly dT tail. Slide-seq technique was applied on 4 different patients—a total of 12 samples: 2 healthy tissues collected from cystoprostatectomy surgeries (patients with bladder cancer), one tumor tissue of gleason 4+3 and one tumor tissue of Gleason 5+4 collected from prostatectomy surgeries along with their adjacent-normal tissues (Table S2). Two samples from each tissue were collected. OCT-embedded frozen tissue samples were warmed to -20°C in a cryostat (Leica CM3050S) and serially sectioned at a 10 µm thickness (2-3 Slide-seq array replicates per sample), with consecutive sections used for hematoxylin and eosin staining. Each tissue section was affixed to an array and moved into a 1.5 mL Eppendorf tube for downstream processing. The sample library was prepared as previously described (13) except with the following modifications. Library amplification before tagmentation for the human samples included an additional two cycles, for a PCR program of 1 cycle of 98°C for 2 min, 4 cycles of 98°C for 20 s, 65°C for 45 s, 72°C for 3 min, 11 cycles of 98°C for 20 s, 67°C for 20 s, 72°C for 3 min, and 1 cycle of 72°C for 5 min. The PCR was performed in a final volume of 200 µL of PCR mix, divided into 4 PCR tubes. Libraries were sequenced using the following read structure on a NovaSeq (S2; Illumina): Read1: 42 bp; Read2: 41 bp; Index1: 8 bp, and sequences were processed as previously described (13) using the pipeline available at <https://github.com/MacoskoLab/slideseq-tools>.”

We used the Broad Institute pipeline (from <https://github.com/MacoskoLab/slideseq-tools.git>) to generate the count-matrices and the bead locations from the raw BCL files.

2. Looking at the map quality in Suppl Table 1, the mean read per cell varies from 2,500 to 200,000. How does this influence data processing and data integration with Conos? Do criteria have been applied to exclude cells or samples?

We thank the reviewer for pointing this out. For the data integration, Conos conducts library size normalization to make the distribution of expression magnitudes comparable. To further measure the result of data integration, we showed the cell annotations in joint UMAP embedding for each sample type. Both low and high mean read samples were well integrated (**Figure R23**).

Figure R23. The analysis integrates scRNA-seq from 37 different tissues: five healthy prostate tissues, 9 adjacent-normal tissues collected from patients with low grade prostate cancer, 5 adjacent-normal tissues collected from patients with high grade prostate cancer, 10 tumor tissues collected from patients with low grade prostate cancer, 8 tumor tissues collected from patients with high grade prostate cancer.

In addition, we did not omit samples based on mean reads per cell but excluded some low-quality cells with the following criteria (1) cells with fewer than 600 total UMI per cell (2) cells with high doublet score above 0.4. We updated the “STAR Method” section to clarify.

Line 981:

“scRNA-seq data processing and analysis

Sequencing data were processed using 10X Cell Ranger with default parameters (version 3.0.1), aligned to GRCh37 human reference genome. In total, we obtained 187,457 cells. We removed cells with less than 600 total UMI considered as “low quality” cells. The obtained read count matrices were further analyzed with Scrublet (107) for doublets identification. Scrublet scores above 0.4 were omitted. After quality control, 179,359 cells from 39 samples were obtained. Two samples from PCA24 (PCA24-N-LG_Collagenases+Dispace, PCA24-N-LG_Rocky (Table S2)) were only used to compare dissociation protocol and they were not added to the comparative analysis. We used Conos (15) ($k=15$, $k.self=5$, $matching.method='mNN'$, $metric='angular'$, $space='PCA'$) to integrate multiple scRNA-seq datasets together. Principal component analysis was performed on 2000 genes with the most variable expression was selected by Conos. Leiden

clustering was used to build to determine joint cell clusters across the entire dataset collection. First 15 principal components were used to perform UMAP embedding.”

3. Line 101: it stands that 179,359 cells have been analyzed. When we sum Table S2-column 2 we find that 187,457 cells analyzed. Can the authors explain?

In total we sequenced 187,457 cells. After quality control, 179,359 cells from 39 samples were obtained. We did not include the two samples (PCA24-N-LG_Collagenases+Dispase, PCA24-N-LG_Rocky) into comparative analysis as the data from these two samples were used only to compare the differences in cell yield and transcriptome profile comparison of two dissociation protocols.

In total, 179,359 cells were used in our analysis. We have updated the paragraph in “STAR Method” section with a detailed description of data pre-processing and quality control. In addition, we added a column in Table S2 to indicate the number of cells after quality control.

Line 981:

“scRNA-seq data processing and analysis

Sequencing data were processed using 10X Cell Ranger with default parameters (version 3.0.1), aligned to GRCh37 human reference genome. In total, we obtained 187,457 cells. We removed cells with less than 600 total UMI considered as “low quality” cells. The obtained read count matrices were further analyzed with Scrublet (107) for doublets identification. Scrublet scores above 0.4 were omitted. After quality control, 179,359 cells from 39 samples were obtained. Two samples from PCA24 (PCA24-N-LG_Collagenases+Dispase, PCA24-N-LG_Rocky (Table S2)) were only used to compare dissociation protocol and they were not added to the comparative analysis. We used Conos (15) (k=15, k.self=5, matching.method='mNN', metric='angular', space='PCA') to integrate multiple scRNA-seq datasets together. Principal component analysis was performed on 2000 genes with the most variable expression was selected by conos. Leiden clustering was used to build to determine joint cell clusters across the entire dataset collection. First 15 principal components were used to perform UMAP embedding.”

4. Why aren't any neutrophils found in the dataset?

This is an important observation. To address this comment, we checked for the expression of common neutrophils marker genes including FCGR3B (21), CEACAM8 (22), ELANE (23) and LCN2 (24). Our dataset showed no expression for these genes in our cell populations (**Figure R24A**) and after zooming-in into the myeloid populations (**Figure R24B**).

Compared to other cell types, mature neutrophils are known to have relatively low RNA content and high levels of RNases resulting in fewer transcripts detected in Gel Bead-In EMulsions (GEMs), and less usable sequencing reads. Based on this, we lowered the threshold for the total UMIs (cutoff of total counts per cell lowered to 300 instead of 600) and checked for the expression of the common neutrophil markers (FCGR3B, CEACAM8, ELANE and LCN2. Our dataset showed no expression for these genes in our cell populations (**Figure R25A**) and after zooming-in into the myeloid populations (**Figure R25B**).

All cells (total UMI/cell= 600)

Myeloid population (total UMI/cell= 600)

Figure R24. UMAP embedding of all cells (A) and myeloid population (B) showing no expression of common neutrophil markers with total UMI/cell equal to 600.

All cells (total UMI/cell= 300)

Myeloid population (total UMI/cell= 300)

Figure R25. UMAP embedding of all cells (A) and myeloid population (B) showing no expression of common neutrophil markers with total UMI/cell equal to 300.

In addition, neutrophils are known to be particularly sensitive to degradation during the process of collection, dissociation, and flow sorting prior to RNA sequencing and thus can be easily ‘lost’ in the purification prior to 10x encapsulation.

We added this point to the “**Discussion**”:

Line 807: “No neutrophils were obtained within our prostate cancer dataset even after lowering the number of reads per cell to 300 instead of 600 (data not shown). Mature neutrophils are known to have relatively low RNA content and high levels of RNases, resulting in fewer transcripts detected in Gel Bead-In EMulsions (GEMs), and less usable sequencing reads. Neutrophils are also particularly sensitive to degradation after collection or during scRNA-seq process, suggesting that we lost the cells due to technical issues.”

5. Even though the SlideSeq data is used mostly as validation, much of the manuscript's novelty is based on these data. Nevertheless, only one patient (n=1) has been analyzed. Increasing the sample size, if in scope, could make observations more solid.

Thank you. The Slide-seq data was collected from 4 patients with a total of 12 samples: 2 healthy tissues collected from cystoprostatectomy surgeries (2 patients with bladder cancer), one tumor tissue of Gleason (4+3) and one tumor tissue of Gleason (5+4) collected from 2 patients underwent prostatectomy surgeries along with their adjacent-normal tissues (please check **Table S2** for details). Two samples from each tissue were collected. The Slide-seq processing and sequencing paragraph in the manuscript has been modified to make it clear regarding the number of samples used (see below).

The focus of our computational analysis is to delineate context dependent and cell-type specific differential expression analysis. To that end, we have taken two samples from the same tissue and used them as replicates. We note that, to analyze the localized differences, we needed to first identify the context (in our case the tumor affected region) and for a lot of patients it was hard to find such a clearly recognizable context as the tumor has infiltrated the entire puck that is used for Slide-seq. In future, we aim to extend our analysis to multiple patients and multiple samples as the data become abundant.

Line 1148:

“Slide-seqV2 processing and sequencing

Slide-seq arrays were prepared and spatial bead barcodes sequenced following Slide-seqV2 (13) protocol using arrays created with custom synthesized barcoded beads

(5’-

TTT_PC_GCCGGTAATACGACTCACTATAGGGCTACACGACGCTCTTCCGATCTJJJJJJ
TCTTCAGCGTTCCCGAGAJJJJJNNNNNNNVVT30-3’)

with a photocleavable linker (PC), a bead barcode sequence (J, 14 bp), a UMI sequence (NNNNNNNVV, 9 bp), and a poly dT tail. Slide-seq technique was applied on 4 different patients- a total of 12 samples: 2 healthy tissues collected from cystoprostatectomy surgeries (patients with bladder cancer), one tumor tissue of gleason 4+3 and one tumor tissue of Gleason 5+4 collected from prostatectomy surgeries along with their adjacent-normal tissues (Table S2). Two samples from each tissue were collected. OCT-embedded frozen tissue samples were warmed to -20°C in a cryostat (Leica CM3050S) and serially sectioned at a 10 µm thickness (2-3 Slide-seq array replicates per sample), with consecutive sections used for hematoxylin and eosin staining. Each

tissue section was affixed to an array and moved into a 1.5 mL Eppendorf tube for downstream processing. The sample library was prepared as previously described (13) except with the following modifications. Library amplification before tagmentation for the human samples included an additional two cycles, for a PCR program of 1 cycle of 98°C for 2 min, 4 cycles of 98°C for 20 s, 65°C for 45 s, 72°C for 3 min, 11 cycles of 98°C for 20 s, 67°C for 20 s, 72°C for 3 min, and 1 cycle of 72°C for 5 min. The PCR was performed in a final volume of 200 µL of PCR mix, divided into 4 PCR tubes. Libraries were sequenced using the following read structure on a NovaSeq (S2; Illumina): Read1: 42 bp; Read2: 41 bp; Index1: 8 bp, and sequences were processed as previously described (13) using the pipeline available at <https://github.com/MacoskoLab/slideseq-tools>.”

6. Reading 'Table S2-Cases_for_Slide_seq' is very confusing: what patient are we talking about? The median read per cell identified per Slide-seq is very low ranging from 400 to 650. The number of reads per cell is almost equal to the number of genes detected. Is that enough to call the cell types?

Thank you. We have added the patient ID from **Table S2** to the figure legends to help to better identify the samples. Compared to other methods, Slide-seqV2 provides relatively lower read depths but a higher number of cells in addition to the important corresponding spatial coordinates. In our study, we benefited from having combined single cell sequencing (10x) and spatial transcriptome to increase our ability to identify cell types even in the setting of the very low total number of reads per cell. In addition, we evaluated cell type specific marker gene expression (**Figure R26**) in spatial data to further confirm the cell identity.

Figure R26. A. (Figure S1G) Dotplot representing key-marker gene expression in major cell types in Slide-seqV2. **B. (Figure 2F)** Dotplot representing key-marker gene expression in epithelial subpopulations in Slide-seqV2. The color represents scaled average expression of marker genes in each cell type, and the size indicates the proportion of cells expressing marker genes.

7. Suppl. Fig. 2C: Only a few samples seem to contribute substantially to the malignant cell population. Could this influence the DEGs identified as 'prostate tumor signature'? Could the signature be useful to discriminate between malignant and non-malignant cells in other malignancies with luminal-epithelial cell origin?

We appreciate this comment. As we mentioned in the manuscript, our dissociation protocol tends to enrich and preserve the immune cells. However, we were still able to obtain the major four epithelial subclusters comparing our dissociation protocol to a different dissociation protocol “Rocky” from Henry et al. (11) (**Figure 27A**) even though the epithelial cell proportion was significantly higher using “Rocky” dissociation protocol (**Figure 27B**). This suggested that we lost epithelial/tumor cells from the different clusters using our protocol, mostly due to technical biases.

However, our experimental design using the three different fractions: healthy, adjacent normal and tumor prostate tissues allowed us to identify prostate tumor signature based on our data analysis explained below in details.

Line 1054

“Generation of the “Prostate Tumor Gene Signature””

To generate a gene expression signature that is clinically relevant, we compared the gene expression profiles between tumor cells and non-tumor luminal cells in tumor fraction. Only the upregulated genes with Z-score > 3 were selected and taken into subsequent analysis. We next screened each of the DEGs based on their expression in healthy prostate tissue, requiring each gene to be expressed in less than 5% cells of all epithelial cells. In total, we identified 8 significant DEGs that

met the above criteria. The average expression of these curated DE genes is regarded as the diagnosis signature score, later used on multiple bulk RNAseq data to quantify the predictive accuracy of such signatures.”

In addition, based on the reviewer’s suggestion, we applied our “Prostate Tumor Gene Signature” on another cancer type of luminal-epithelial cell origin such as breast cancer. To this end, the average expression of the curated “Prostate Tumor Gene Signature” used on bulk RNA-seq data of breast cancer (GSE70770). ROC analysis did not show a strong predictive ability to distinguish tumor samples from adjacent normal samples with an AUC score of 0.63 (**Figure R28A**).

Also, we examined the expression of the 8 genes of our “Prostate Tumor Gene Signature” on single cell RNA-seq data generated from pre-treated human breast cancer where all major cell types were represented across all tumors (Wu et al., Nat Genet. 2021). The aim was to examine whether there are differences in their expression level comparing different tumor and luminal subsets collected from breast tumor samples at the single cell level. Data did not show any significant differences in the gene expression comparing the tumor to luminal subsets (**Figure R28B**) with no expression for prostate cancer antigen 3 (PCA3) in any of the different subsets as it is specific for prostate cancer.

We added a small paragraph to our manuscript (**Discussion**) mentioning the specificity of the “Prostate Tumor Gene Signature” to prostate cancer based on the following comment and data obtained (**Figure 28**):

Line 816:

“The “Prostate Tumor Gene Signature” showed specificity to prostate cancer as it did not show a strong differentiation between normal and tumor samples when applied on breast cancer data as another cancer type of luminal-epithelial cell origin (data not shown).”

Figure R27. A. Joint embedding represents the detailed annotation of epithelial subpopulations in prostate tissues collected from Henry et al. paper using “Rocky” dissociation protocol (left) and from our dataset using “Collagenases+Dispase” dissociation protocol (right). **B. (Figure 1D)** Stacked barplots showing the fractional composition of cell number for different cell populations within scRNA-seq (using two different dissociation protocols: Collagenases+Dispase and Rocky) and Slide-seqV2.

Figure R28. **A.** ROC curves for “Prostate Tumor Gene Signature” score applied on breast cancer dataset (GSE70770). **B.** Violin plot showing the expression of the 8 genes of “Prostate Tumor Gene Signature” in different tumor and luminal subsets of breast tumor samples.

8. Line 169-174: the authors applied ICA analysis and obtained Figure S3A. It is really not clear how this analysis was applied and why only 6 samples have been selected. The cluster C1 shows a signature with FOS, JUNB. These genes come along with tissue dissociation artifacts as abundantly described before. Is it possible that tissue dissociation induces spurious gene signatures in addition to the tumor signature?

Thank you. As we mentioned in the comment above (comment# 7), our dissociation protocol tends to enrich and preserve the immune cells. Only 6 samples were predicted with abundant tumor cells (>40 cells) (**Figure 29A**). To examine the tumor cells heterogeneity, we extracted tumor cells and re-ran data integration using Conos. In short, each individual dataset was first normalized and projected to low dimension space using ICA. Different samples were then aligned together using Conos. UMAP embedding was estimated using default parameter settings. Leiden clustering (conos:findCommunities) was used to determine joint cell clusters across the entire dataset collection. We identified three tumor cell sub-clusters (C1-C3) (**Figure R29B**). Cluster 1 exhibits a different expression profile, with high expression of JUNB, FOS, KLF6, EGR1, and IER2. EGR1 was reported to promote the progression of prostate cancer (25) and IER2 can promote tumor cell motility and metastasis (26) (**Figure 29C**).

Figure R29. **A.** Barplot represents the number of tumor cells in prostate tumor samples of different patients. **B.** UMAP embedding of the three different clusters of tumor cells. **C.** (Figure S3A) Heatmap showing differentially expressed (DE) genes in the three malignant cell subclusters.

To check whether FOS and JUNB expression is an artifact of tissue dissociation, we looked for their expression in our Slide-seq spatial single cell data where no dissociation took place. FOS and JUNB were highly expressed in some of the tumor cells, similarly to KLF6 and EGR1 which also showed high expression in cluster 1 (Figure R29C, Figure R30A). Furthermore, we extracted tumor cells and performed dimension reduction with tSNE. Data showed a cluster of tumor cells with high expression of FOS, JUNB, KLF6 and EGR1 (Figure 30B). Thus, we do not believe that FOS and JUNB expression is an artifact of our tissue dissociation.

Figure R30. A. Spatial presentation of epithelial subpopulations (top) and expression for FOS, JUNB, KLF6 and EGR1 in tumor cells collected from a tumor (HG) Slide-SeqV2 pucks (Tumor-02 (HG)). **B.** tSNE visualization of FOS, JUNB, KLF6 and EGR1 expression in tumor cells collected from a high-grade (Tumor-02 (HG)) spatial sample.

9. Figure 2H: Regarding the approach of linear regression to correct for context-dependent admixture : The authors do not show if their approach confers an advantage over only taking high confidence "pure" beads into account.

We thank the reviewer for this observation. For a deconvolution tool (e.g., RCTD), inferring the identity of mixed cell-types is the main goal and such tools take information from the "pure bead". In our case, we started from that RCTD output. Unfortunately, even when we take the pure beads that are marked as singlets (by RCTD), they do show to have mixed transcriptomic material from neighboring tumor cells where data showed the over-expression of tumor signature genes in "non"-tumor cell-types (**Figure 31**). This observation led us to believe that there has been a linear mixture throughout a context (in our case characterized by over-abundance of tumor cells). Therefore, we did the regression-based correction at the level of pseudo-bulk data rather than bead level one. Please note that if we want to correct the individual beads, we do need to use the pure beads, but in this case that was not the goal. Additionally, if we only consider pure beads and remove the mixed (with uncertain cell-types), we lose almost 14.2% of the beads decreasing the depth/power of our analysis.

Figure 31. Violin plot showing tumor/epithelial key marker genes' expression in stromal subsets, comparing Slide-seq "pure bead" data (Tumor-02 HG) to 10X single cell data (PCA3).

10. Cell population abundance comparison between sample types should take into account the proportional nature of population counts. Therefore, in addition to or instead of Wilcoxon more specialized statistical models should be used, for example (10.1038/s41467-021-27150-6). Especially for M2 macrophages in this manuscript, this would be important. As is now, it could also be possible that antigen presenting macrophages are reduced in Adj-N and Tumor samples and the statistical test used finds M2-macrophages to be enriched, even though this could be an artifact of negative correlation.

In simple proportion tests, the proportions of all cell types are not independent as a significant expansion of one cell type can lead to a significant decrease in the proportion of other cell types. Due to high patient-to-patient variability, we only obtain few significant differences in cell abundance. M2-macrophages was the one we highlighted and validated in our manuscript. As the reviewer suggested, we tried both CoDA (Petukhov V. et al. BioRxiv 2022) and scCODA (27) for cell compositional analysis. The CoDA analysis avoids the non-independence of cell type fractions and translates the abundances of N cell types into N-1 simplex space using isometric log-ratio transformation (ilr). CoDA was implemented in the cacao package (<https://github.com/kharchenkolab/cacao>). CoDA analysis showed that macrophage3 (which represents M2-macrophages) was significantly enriched in the tumor fraction compared to the healthy fraction (**Figure R32**). We also tried scCODA but we did not get a significant increase of M2-macrophages.

Figure R32. Changes in cell composition evaluated by Compositional Data Analysis (CoDA). The x-axis indicates the separating coefficient for each cell type with the positive values corresponding to increased abundance in the tumor, and negative values corresponding to decreased abundance in the tumor. Red line indicates p value =0.05.

In addition, compositional differences could also be presented by cluster-free cell compositional analysis. Cluster-free analysis of compositional difference does not rely on cell types identification and increases the resolution of compositional shift. In the common UMAP embedding space, we first compute kernel density in joint embedding space for each sample (bin = 400). Obtained density matrix was normalized by quantile normalization. The average density of each sample group was shown in **Figure R33A**. To measure the differential cell density between sample groups, we performed a wilcoxon rank sum test between sample groups in each girded bin. As **Figure R33A** shows, Macrophage3 cluster representing M2-macrophages have higher z-score, suggesting an expansion for the following subset (M2-macrophages) in tumor fraction compared to healthy fraction.

More importantly, we also validated the higher M2-macrophages infiltration in tumor tissues compared to their matched adjacent-normal tissue using multiplexed Immunohistochemistry and this was more pronounced in cases of high Gleason scores (**Figure 33B**).

A.

B.

Figure R33. A. Changes in the composition of all compartments combining all sample fractions and is visualized as cell density on the joint embedding. Statistical assessment of the cell density differences comparing tumor with healthy. A two-side Wilcoxon test was used, visualized as a Z score. Red indicates increased cell abundance in the tumor, blue indicates decreased cell abundance in the tumor. **B.** Quantification of absolute number of M2-macrophages from mIHC data comparing tumor tissues to their matched adj-normal tissues collected from prostatectomy cases of different Gleason scores.

Petukhov V. et al., Case-control analysis of single-cell RNA-seq studies. doi: <https://doi.org/10.1101/2022.03.15.484475>.

11. The authors use various gene signatures to calculate signature scores. Where did the authors derive those signatures from?

The references for the various signatures have been added to **Table S4**.

12. In lines 349-351 the authors state that they assayed Treg activity as a surrogate for CD4+ T-cell function. As Tregs and other CD4+ T-cell subpopulations (e.g. CD4+ effector memory T-cells) have very different functions, Treg activity seems not like an appropriate surrogate for overall CD4+ T-cell function. Furthermore, Treg activity was not assayed. A score based on a gene signature was calculated, but no functional assay was conducted. Please also refer to 'Treg activity score' in Figures 6F and 6G to avoid confusion.

We appreciate the Reviewer's suggestion, and we agree that we need to refer to "Treg activity score" in Figures 6F and 6G. The figures and figure legends have been modified.

Regarding the text, we apologize for the confusion as we did not mean the overall CD4+ T-cell function but a part of CD4+ T cell function related to only Tregs. The relevant text has been revised:

Line 520: "In Tregs subpopulation, we checked for Treg activity gene expression (Treg activity score) (90,91) (**Table S4**). Data showed significantly higher Treg activity score in Tregs collected from the tumor compared to adj-normal and healthy samples (Figure 6F)."

Minor comments:

1. Figure 1a can be enhanced to add a description of the cohort (how many patients, how many patients analyzed with scRNA-seq, how many with slide-seq, etc...).

We have added the number of samples used for scRNA-seq and Slide-seq as Adjacent-normal, Tumor and Healthy next to each sample type.

2. Figure 1D: the color code should be defined in the legend

The colors have been corrected and a figure legend has been added.

Figure R34. (Figure 1D) Stacked barplots showing the fractional composition of cell number for different clusters within scRNA-seq (using two different dissociation protocols: Collagenases+Dispase and Rocky) and Slide-seqV2.

3. Figure 3D: Please add a legend indicating that red circles are tumor markers to make it easier visible what they mean (even though it is in the legend and supplementary info).

A legend indicating that red circles are tumor markers has been added (**Figure 3D**).

Figure R35. (Figure 3D) The scatterplot showing the effect of linear model-based correction on Endothelial-2 cells. Red dots indicate tumor marker genes. The x-axis is the log-fold change of the genes without the correction, the y-axis is the same after the correction. The top DE genes are text-labeled.

4. Typo: Figure 3G: Please correct 'Spatial Auto-correction' to 'Spatial Auto-Correlation'

The typo has been corrected.

5. Typo: Supplementary Figure 5B: it is 'pseudotime' not 'sudotime'

The typo has been corrected.

References

1. Peng J, Sun B-F, Chen C-Y, Zhou J-Y, Chen Y-S, Chen H, et al. Single-cell RNA-seq highlights intra-tumoral heterogeneity and malignant progression in pancreatic ductal adenocarcinoma. *Cell Res.* 2019;29:725–38.
2. Cillo AR, Kürten CHL, Tabib T, Qi Z, Onkar S, Wang T, et al. Immune Landscape of Viral- and Carcinogen-Driven Head and Neck Cancer. *Immunity.* 2020;52:183-199.e9.
3. Zhang Q, He Y, Luo N, Patel SJ, Han Y, Gao R, et al. Landscape and Dynamics of Single Immune Cells in Hepatocellular Carcinoma. *Cell.* 2019;179:829-845.e20.
4. Lambrechts D, Wauters E, Boeckx B, Aibar S, Nittner D, Burton O, et al. Phenotype molding of stromal cells in the lung tumor microenvironment. *Nat Med.* 2018;24:1277–89.
5. Baryawno N, Przybylski D, Kowalczyk MS, Kfoury Y, Severe N, Gustafsson K, et al. A Cellular Taxonomy of the Bone Marrow Stroma in Homeostasis and Leukemia. *Cell.* 2019;177:1915-1932.e16.
6. Nehls V, Denzer K, Drenckhahn D. Pericyte involvement in capillary sprouting during angiogenesis in situ. *Cell Tissue Res.* 1992;270:469–74.
7. Rippe C, Morén B, Liu L, Stenkula KG, Mustaniemi J, Wennström M, et al. NG2/CSPG4, CD146/MCAM and VAP1/AOC3 are regulated by myocardin-related transcription factors in smooth muscle cells. *Sci Rep-uk.* 2021;11:5955.
8. Liu R, Jin J-P. Calponin isoforms CNN1, CNN2 and CNN3: Regulators for actin cytoskeleton functions in smooth muscle and non-muscle cells. *Gene.* 2016;585:143–53.
9. Duband J, Gimona M, Scatena M, Sartore S, Small JV. Calponin and SM22 as differentiation markers of smooth muscle: spatiotemporal distribution during avian embryonic development. *Differentiation.* 1993;55:1–11.
10. Irshad S, Bansal M, Castillo-Martin M, Zheng T, Aytes A, Wenske S, et al. A Molecular Signature Predictive of Indolent Prostate Cancer. *Sci Transl Med.* 2013;5:202ra122.
11. Henry GH, Malewska A, Joseph DB, Malladi VS, Lee J, Torrealba J, et al. A Cellular Anatomy of the Normal Adult Human Prostate and Prostatic Urethra. *Cell Reports.* 2018;25:3530-3542.e5.
12. Team TCIGC, O’Flanagan CH, Campbell KR, Zhang AW, Kabeer F, Lim JLP, et al. Dissociation of solid tumor tissues with cold active protease for single-cell RNA-seq minimizes conserved collagenase-associated stress responses. *Genome Biol.* 2019;20:210.
13. Halse H, Colebatch AJ, Petrone P, Henderson MA, Mills JK, Snow H, et al. Multiplex immunohistochemistry accurately defines the immune context of metastatic melanoma. *Sci Rep-uk.* 2018;8:11158.
14. Boisson A, Noël G, Saiselet M, Rodrigues-Vitória J, Thomas N, Fontsa ML, et al. Fluorescent Multiplex Immunohistochemistry Coupled With Other State-Of-The-Art Techniques to Systematically Characterize the Tumor Immune Microenvironment. *Frontiers Mol Biosci.* 2021;8:673042.

15. Chen Z, Zhou L, Liu L, Hou Y, Xiong M, Yang Y, et al. Single-cell RNA sequencing highlights the role of inflammatory cancer-associated fibroblasts in bladder urothelial carcinoma. *Nat Commun.* 2020;11:5077.
16. Bartoschek M, Oskolkov N, Bocci M, Lövrot J, Larsson C, Sommarin M, et al. Spatially and functionally distinct subclasses of breast cancer-associated fibroblasts revealed by single cell RNA sequencing. *Nat Commun.* 2018;9:5150.
17. Ramos RN, Missolo-Koussou Y, Gerber-Ferder Y, Bromley CP, Bugatti M, Núñez NG, et al. Tissue-resident FOLR2+ macrophages associate with CD8+ T cell infiltration in human breast cancer. *Cell.* 2022;185:1189-1207.e25.
18. Pauken KE, Sammons MA, Odorizzi PM, Manne S, Godec J, Khan O, et al. Epigenetic stability of exhausted T cells limits durability of reinvigoration by PD-1 blockade. *Science.* 2016;354:1160–5.
19. Blackburn SD, Shin H, Freeman GJ, Wherry EJ. Selective expansion of a subset of exhausted CD8 T cells by α PD-L1 blockade. *Proc National Acad Sci.* 2008;105:15016–21.
20. Kfoury Y, Baryawno N, Severe N, Mei S, Gustafsson K, Hirz T, et al. Human prostate cancer bone metastases have an actionable immunosuppressive microenvironment. *Cancer Cell.* 2021;39:1464-1478.e8.
21. Wang Y, Jönsson F. Expression, Role, and Regulation of Neutrophil Fc γ Receptors. *Front Immunol.* 2019;10:1958.
22. Rayes RF, Vourtzoumis P, Rjeily MB, Seth R, Bourdeau F, Giannias B, et al. Neutrophil Extracellular Trap–Associated CEACAM1 as a Putative Therapeutic Target to Prevent Metastatic Progression of Colon Carcinoma. *J Immunol.* 2020;204:2285–94.
23. Cui C, Chakraborty K, Tang XA, Zhou G, Schoenfelt KQ, Becker KM, et al. Neutrophil elastase selectively kills cancer cells and attenuates tumorigenesis. *Cell.* 2021;184:3163-3177.e21.
24. Xie X, Shi Q, Wu P, Zhang X, Kambara H, Su J, et al. Single-cell transcriptome profiling reveals neutrophil heterogeneity in homeostasis and infection. *Nat Immunol.* 2020;21:1119–33.
25. Li L, Ameri AH, Wang S, Jansson KH, Casey OM, Yang Q, et al. EGR1 regulates angiogenic and osteoclastogenic factors in prostate cancer and promotes metastasis. *Oncogene.* 2019;38:6241–55.
26. Neeb A, Wallbaum S, Novac N, Dukovic-Schulze S, Scholl I, Schreiber C, et al. The immediate early gene *Ier2* promotes tumor cell motility and metastasis, and predicts poor survival of colorectal cancer patients. *Oncogene.* 2012;31:3796–806.
27. Büttner M, Ostner J, Müller CL, Theis FJ, Schubert B. scCODA is a Bayesian model for compositional single-cell data analysis. *Nat Commun.* 2021;12:6876.

REVIEWERS' COMMENTS

Reviewer #2 (Remarks to the Author):

Responses and revisions are comprehensive and sufficiently address prior comments. I feel it is now suitable for publication.

Reviewer #3 (Remarks to the Author):

The authors have answered all my concerns.

Reviewer #4 (Remarks to the Author):

Authors have done extensive work in their revision. Additional data from other tumor types with the new analysis are satisfactory and address the raised concerns properly.

Data availability statement should be updated to include accession numbers also to raw data as currently these are not available through the GEO accession link provided.